# Assessing processing-based measures of implicit statistical learning: Three serial reaction time experiments do not reveal artificial grammar learning

Holly E. Jenkins[1], Phyllis Leung[2], Faye Smith[3], Nick Riches[3], Benjamin Wilson[4,5]*

1 Department of Education, University of Oxford, Oxford, United Kingdom, 2 Derbyshire Healthcare NHS Foundation Trust, Matlock, United Kingdom, 3 School of Education, Communication and Language Sciences, Newcastle University, Newcastle Upon Tyne, United Kingdom, 4 Department of Psychology, Emory University, Atlanta, Georgia, United States of America, 5 Emory National Primate Research Center, Emory University, Atlanta, Georgia, United States of America

* benjamin.wilson@emory.edu

**Data Availability Statement:** The data and analysis are available on OSF (DOI 10.17605/OSF.IO/DETP3).

## Abstract

Implicit statistical learning, whereby predictable relationships between stimuli are detected without conscious awareness, is important for language acquisition. However, while this process is putatively implicit, it is often assessed using measures that require explicit reflection and conscious decision making. Here, we conducted three experiments combining an artificial grammar learning paradigm with a serial reaction time (SRT-AGL) task, to measure statistical learning of adjacent and nonadjacent dependencies implicitly, without conscious decision making. Participants viewed an array of six visual stimuli and were presented with a sequence of three auditory (nonsense words, Expt. 1; names of familiar objects, Expt. 2) or visual (abstract shapes, Expt. 3) cues and were asked to click on the corresponding visual stimulus as quickly as possible. In each experiment, the final stimulus in the sequence was predictable based on items earlier in the sequence. Faster responses to this predictable final stimulus compared to unpredictable stimuli would provide evidence of implicit statistical learning, without requiring explicit decision making or conscious reflection. Despite previous positive results (Christiansen et al. 2009 and Misyak et al. 2010) we saw little evidence of implicit statistical learning in any of the experiments, suggesting that in this case, these SRT-AGL tasks were not an effective measure implicit statistical learning.

## Introduction

The environment in which we live is richly structured; events do not occur randomly, but some events predict others. For example, rain can typically be predicted by the presence of dark clouds, or the sound of a car horn may predict a traffic accident. We often learn about this structure via implicit statistical learning, whereby knowledge is acquired automatically, without explicit awareness of the relationships between objects, events or actions. This same

**Funding:** The research was funded by a Sir Henry Wellcome Fellowship (BW: WT110198/Z/15/Z; https://wellcome.org/) and by ORIP/P51OD011132 (NIH). The funders had no role in study design, data collection and analysis, decision to publish, or preparation of the manuscript.

**Competing interests:** This does not alter our adherence to PLOS ONE policies on sharing data and materials.

process of acquiring structure based on statistical cues is also central to language learning, where grammatical rules are often acquired implicitly and without conscious awareness [1–3].

In a series of seminal studies, Saffran and colleagues showed that infants learn to recognise word boundaries in a continuous speech stream based only on statistical regularities from both artificial stimuli [4–7] and in natural languages [8]. Beyond detecting word boundaries, similar processes are assumed to be involved in implicitly learning grammatical relationships between words or phrases [9–13]. These artificial grammar learning experiments can be used to assess the learning of both relationships between adjacent words in a sentence, and also more complex nonadjacent dependencies [14, 15]. These types of tasks have provided valuable insights into how grammatical knowledge can be acquired based on the statistical regularities in the input to which we are exposed, without conscious awareness of these grammatical rules [12].

Artificial grammar learning studies typically consist of two phases. First is an exposure or learning phase in which a participant is presented with many sequences of stimuli (normally auditory or visual), which conform to certain rules and thus contain predictable statistical regularities (e.g., some stimuli co-occur more frequently than others). In this phase participants might be asked to attend to the stimuli but are usually not told to search for any particular rules or patterns, only to passively view or listen to the stimuli. In a subsequent testing phase, participants are presented with some sequences which follow the same rules as the exposure sequences that they previously heard (grammatical sequences) and some sequences which violate these rules (ungrammatical sequences). Different responses to the grammatical sequences compared to the ungrammatical sequences can be taken as evidence that the participant learned at least some of the rules or statistical regularities during the initial exposure phase. The inability to identify any of the rules or relationships within the grammar suggests that this knowledge is implicit [12, 16, 17].

The testing phase of artificial grammar learning experiments often rely on methods that, at least to some extent, assess explicit knowledge. In these studies, following exposure, participants typically complete a grammaticality judgement task, in which they are told that the sequences that they have just heard followed certain rules or patterns. They are then told that they must indicate whether the testing sequences that they are about to hear are "well-formed", in other words, whether they follow the same or different patterns [7, 12]. These 'reflection-based' methods require participants to consciously think about what they have learned and make explicit decisions about the grammaticality of a sequence. This may not be an appropriate way of measuring implicit statistical learning, as these tasks are potentially tapping into other, more explicit decision-making processes [18]. To avoid this, some studies have used 'processing-based' measures of learning, which measure other variables that are facilitated by implicit statistical learning, and avoid any interference from explicit decision-making processes [2, 18–23].

One processing-based approach that avoids relying on explicit decision-making uses reaction times to measure implicit learning. One of the reasons that it is beneficial to learn statistical regularities in our environment is because it allows us to better predict future events, and thus react faster to them. Serial Reaction Time (SRT) tasks have demonstrated implicit learning by showing that if participants are asked to make sequences of motor responses (e.g., touch stimuli in different spatial locations), they become faster at producing predictable sequences than random sequences [24–26]. In these tasks, faster reaction times demonstrate implicit learning of action sequences.

The learning observed in traditional SRT tasks may be somewhat different to the statistical learning found in artificial grammar learning tasks and indeed in natural language, in which participants must learn dependencies between sensory stimuli rather than between spatial

locations. Moreover, traditional SRT tasks are thought to rely on different cognitive and neural mechanisms than artificial grammar learning tasks, for example procedural memory [27]. However, SRT tasks have been adapted to better assess artificial grammar learning, and to measure the learning of associations between stimuli, rather than the relationships between specific spatial locations. For example, Misyak et al. [20] presented participants with a visual array containing two rows of three nonsense word stimuli (see Fig 1 for a similar design). Participants were then presented with a sequence of three auditory nonsense words, each corresponding to one of the nonsense words on screen and asked to click on the matching nonsense word stimulus as quickly as possible. Unbeknownst to the participants, the first nonsense word of the sequence always predicted the final nonsense word, by way of a nonadjacent dependency. Over the course of the experiment participants became faster at responding to this final predictable nonsense word, and reaction times were slower in a 'Testing Block' when they were presented with sequences that did not contain the predictable relationships [19] (although see [28–30] for other instances of SRT-like measures for SL, and [31], for an early adaptation of the SRT paradigm). Importantly, in this task although the nonsense word sequences always unfolded from left to right across the screen, the vertical position of the stimuli was randomised, and was not predictable based on the previous nonsense words. This removes the reliance on motor learning of action sequences, and required that the participants learn relationships between the nonsense words themselves.

There has been much discussion relating to the types of processing that are involved in implicit statistical learning tasks, and whether these tasks are effective and reliable measures [32]. Indeed, many tasks measuring implicit statistical learning rely on both implicit and explicit processing, and separating these processes presents challenges [33]. Indeed, a related, although not wholly analogous, line of research has used a range of measures (including subjective confidence ratings, gambling tasks and target detection paradigms) to ask whether the knowledge acquired in these tasks may actually be available to explicit, conscious awareness, and therefore may not truly be 'implicit' knowledge (see [34–39]). This is undoubtedly an important question, although it is somewhat different to the goal of the current study. Here, rather than asking if the learning that is observed is explicitly available, we instead ask whether it is possible to elicit and measure learning without requiring any conscious reflection. One of the key benefits of combining SRT and AGL paradigms over other 'implicit' paradigms (e.g., [39]) is that these tasks retain the flexibility of traditional AGL paradigms, in that they can be used to measure learning of a variety of types of grammatical dependencies, from simpler adjacent dependencies to more complex hierarchical relationships.

Although it was beyond the scope of the current experiments, the long-term aim of this project was to design a task that could be used in the future to assess individual differences in implicit statistical learning in individuals with language and reading difficulties, such as dyslexia. To achieve this goal, we first aimed to broadly replicate the original results ([20]; Experiment. 1), then to design an SRT-AGL task more suitable for those with language difficulties or disorders by removing the requirement to read nonsense words (Experiment 2) or respond to phonological stimuli (Experiment 3), the processing of which is thought to be impaired in dyslexia [40].

Following the SRT-AGL task, participants completed reflection-based tasks to investigate more explicit awareness of the dependencies within the sequences. In Experiments 1 and 2, participants completed a 'Sequence Completion' task and a 'Grammaticality Judgement' task, and in Experiment 3 they also completed a novel 'Sequence Generation' task. In the Sequence Completion task, participants were presented with the first two stimuli in a sequence, as in the SRT-AGL task. However, the final stimulus was omitted, and the participant was asked to complete the sequence without the cue, by selecting the appropriate final stimulus based on

## A. Serial Reaction Time-Artificial Grammar Learning Task

## B. Sequence Completion Task

## C. Grammaticality Judgement Task

**Fig 1. Trial design for Experiment 1.** A) SRT-AGL task. On each trial participants were presented with an array of 6 visual nonsense word stimuli. Every trial contained 2 'A' elements in the left column, 2 'X' elements in the middle column, and 2 'B' items in the right column. Participants were presented with an auditory cue corresponding to one of the two visual stimuli in each column, and were asked to click on the matching visual stimulus as quickly as possible (see Methods). Following their final response, participants were given a score corresponding to their total reaction time across the whole sequence. B) Sequence Completion task. As in the SRT-AGL task, participants were presented with a visual array of stimuli and responded to the first two auditory cue stimuli. However, the final auditory cue stimulus was replaced with a 'beep' sound, and the participants were asked to guess the nonsense word stimulus that they felt completed the sequence. C) Grammaticality Judgement Task. In this task participants were shown a blue fixation spot and were presented with an auditory sequence that was either 'grammatical' or 'ungrammatical'. When the spot changed colour, they pressed one of two keys on the keyboard to indicate whether the sequence followed the same pattern as the sequence they had heard previously or not.

what they had learned during the experiment. In this task, rather than using reaction times as a processing-based measure of learning, response accuracy in selecting this final element was analysed to assess learning that might be somewhat more explicit, and based on conscious reflection. The Grammaticality Judgement task was very similar to those used in previous experiments [19, 20]. Participants were presented with a sequence and explicitly asked whether the sequence followed the same pattern as the sequences they had previously been exposed to, requiring further explicit reflection on their knowledge of the underlying structure of the sequences. In the Sequence Generation task in Experiment 3, participants were tasked with creating their own sequences, without any cues, based on what they had learned. The Sequence Generation task was designed to provide more information regarding the extent to which participants are consciously aware of any implicitly acquired knowledge, as the ability to generate sequences relies on gaining explicit access to knowledge of the dependencies [41, 42]. Taken together, these tasks were included to assess the extent to which participants had access to the information they had (presumably) implicitly learned, in order to make more explicit decisions.

If SRT-AGL tasks are an effective measure of implicit statistical learning, then we would predict that participants would show faster reaction times to the predictable sequences than to unpredictable sequences. If this learning was consciously accessible, we would also predict that those participants who showed implicit learning during the SRT-AGL tasks should also show above chance performance during the subsequent, more explicit tasks. By contrast, if implicitly learned knowledge is inaccessible for immediate conscious decision-making, we would predict no correlation between the SRT-AGL task and the explicit tasks [43, 44].

## General methods

### Stimuli

In three experiments, we used an artificial grammar to assess whether participants implicitly learn nonadjacent dependencies [20, 45]. This grammar generates sequences of three elements of the form 'AXB', where the initial 'A' element (e.g., '$A_1$') predicts the final 'B' element ('$B_1$') forming a nonadjacent dependency ('$A_1XB_1$') while the intervening 'X' element is not dependent on either the 'A' or the 'B' stimuli. In Experiment 3, we also assessed adjacent dependency learning using a similar grammar of the form 'XAB', where the 'A' elements still predicted the 'B' elements, but these items occurred adjacent to one another in a sequence, rather than being separated by the intervening 'X' element.

### Procedure

We conducted three experiments based on previous Serial Reaction Time-Artificial Grammar Learning (SRT-AGL) experiments [19, 20, 46]. All three experiments shared the same general procedure, including an SRT-AGL task, followed by a Sequence Completion task and a Grammaticality Judgement task (see below for details). All participants provided informed consent before taking part.

All experiments took place in testing labs either within the Institute of Neuroscience at Newcastle University or the Department of Psychology at Emory University. Participants were seated approximately 60cm in front of a 24 inch computer monitor (screen resolution 1920*1200 pixels). All experiments were coded using Matlab and PsychToolbox. Responses were made either with the mouse (in the SRT-AGL and Sequence Completion tasks) or by pressing one of two keys on the keyboard (in the Grammaticality Judgement tasks, see below). All experiments had ethical approval from the Newcastle University Human Ethics Committee

for Faculty of Medical Sciences or Emory University Institutional Review Board, and all the participants provided informed consent before taking part in the experiment.

## Serial Reaction Time-Artificial Grammar Learning task

In the SRT-AGL task, participants were presented with an array of 6 visual stimuli arranged in two rows of three images (Fig 1). The vertical position of the stimuli (upper or lower) was pseudo-randomised so that all items occurred equally frequently in the upper and lower positions, and so that the matching 'A' and 'B' elements occurred in the same row 50% of the time, so the correct responses could not be predicted based on the position of the stimuli. 250ms after the visual stimuli appeared, we presented a cue stimulus which corresponded to one of the two visual stimuli in the left-hand column on the screen (see Fig 1, and below for details of stimuli for each experiment) and the participant was instructed to click on the matching stimulus as quickly as possible. Immediately following their response, a cue stimulus corresponding to one of the visual stimuli in the middle column was presented, and again participants reacted as quickly as possible, before a cue corresponding to a stimulus in the right column was presented in the same way. At the end of each trial, after the participant made their final response, participants were given feedback based on their reaction time (see below). The mouse cursor remained in the control of the participant throughout the experiment.

In each experiment the final element of the sequence was predictable based on the previous elements in the sequence (see Stimuli). Therefore, if participants had learned the dependency, they should be able to respond more quickly to this predictable element than the preceding, unpredictable stimuli [19, 20]. Therefore, we calculated the difference in reaction time between the first ($RT_A$) and last element ($RT_B$). This difference ($RT_{A-B}$) gives a measure of the speed increase provided by the predictable stimulus. We also calculated a similar measure of reaction time based on the unpredictable 'X' element ($RT_{X-B}$). The results of these analyses did not differ from those using $RT_{A-B}$, (see S1 Fig in S1 File), so we primarily report the $RT_{A-B}$ results.

In each experiment participants were also presented with ungrammatical sequences in which the final 'B' element in the sequence did not correspond to the 'A' element (e.g., '$A_1X\underline{\textbf{B}_\textbf{2}}$'). In Experiments 1 and 2 these ungrammatical sequences were all presented in a single 'Testing Block' of 24 trials towards the end of the experiment, while in Experiment 3 we used an oddball design, in which the ungrammatical sequences were infrequently presented throughout the experiment (see individual experiment Methods, below). In both cases, we predicted that if participants had learned the dependencies (e.g., $A_1 \rightarrow B_1$), they should implicitly anticipate the predictable 'B' stimulus and therefore when they are asked to click on an unexpected element (e.g., $B_2$) they should show slower reaction times.

## Sequence Completion task

Following the SRT-AGL task each participant took part in a Sequence Completion task consisting of 12 trials. Each trial in this task began identically to the SRT-AGL task: the participants were presented with an array of 6 visual stimuli, after which the first two cue stimuli were presented and participants responded by clicking on the matching stimuli, as before. However, instead of presenting the final cue stimulus (corresponding to the final element in the sequence), participants were presented with either a 'beep' sound (Experiment 1 and 2) or a question mark (Experiment 3) in place of the final cue stimulus (see Fig 1). They were instructed to choose which of the two possible final stimuli they thought completed the sequence. We predicted that if the participants had learned the sequence dependencies in the earlier part of the task, they should be more likely to select the correct stimulus. As the participants were asked to make a conscious decision in this task, this may rely on more explicit

processes, similar to grammaticality judgement tasks. However, unlike the grammaticality judgement tasks, the participants were not informed that the sequences followed any patterns or rules within the sequences prior to this task, and therefore this might offer a more implicit alternative to the grammaticality judgement task.

### Grammaticality Judgement task

We then conducted a more traditional grammaticality judgement task. In this task, participants were presented with 24 sequences of three stimuli (drawn from the SRT-AGL stimuli, with each stimulus (e.g., $A_1$, $A_2$, $A_3$), presented equally frequently) and asked whether or not these sequences 'followed the same pattern' as the sequences they had heard or seen previously, by pressing one of two keys on the keyboard. They were told that if they were not certain they should respond based on their gut feeling. This approach is similar to many other AGL studies, particularly testing adult participants [12].

### Data analysis

In all SRT-AGL task trials, if participants made an error the trial was omitted from the reaction time analysis (<4% of trials contained an incorrect response). We predicted that participants may also make more errors in their selection of the final 'B' elements for the ungrammatical sequences than for the grammatical sequences, although analysis of the data from all three tasks showed that errors were minimal. To remove outliers, we also omitted trials with reaction times that exceeded the mean reaction time + 3SDs, for each participant.

## Experiment 1: Nonsense word SRT-AGL task

### Participants

Twenty-eight adult participants (17 female, 11 male; ages 19–52 years, mean age: 31 years) were recruited using the Institute of Neuroscience participant pool at Newcastle University. All participants had normal or corrected to normal hearing and vision.

### Stimuli

In this experiment the stimuli were monosyllabic consonant-vowel-consonant (CVC) nonsense words (e.g., 'bek', 'kiv', 'jat'; see Table 1), presented in the form 'AXB'. We used 3 'A' and 3 'B' elements, which consistently co-occurred with one another (e.g., '$A_1XB_1$'; if a sequence began with 'bek' it always ended with 'jat'). We included 24 'X' elements (see Table 1), based on the finding that more variability in these uninformative stimuli facilitates the learning of nonadjacent dependencies [45]. The 'X' elements used in this experiment were monosyllabic

**Table 1. Stimuli for Experiment 1.** Sequences all took the form 'AXB' (e.g., *'bek kiv jat'*), in which the 'A' and 'B' elements always co-occurred. There were 3 'A' and 'B' elements and 24 possible 'X' elements giving a total of 72 possible grammatical sequences. Each sequence was presented once in each Learning Block and in the Recovery Block (see Methods). In the Testing Block, ungrammatical pairs of A and B elements (i.e., $A_1\_B_2$, $A_1\_B_3$, $A_2\_B_1$, $A_2\_B_3$, $A_3\_B_1$, $A_3\_B_2$) were presented four times and each 'X' element was presented once.

| 'A' Elements | | 'B' Elements | 'X' elements | | | |
|---|---|---|---|---|---|---|
| *bek* | - | *jat* | *biz* | *heb* | *lod* | *rik* |
| | | | *dil* | *jup* | *lun* | *ruj* |
| *hix* | - | *tef* | *fal* | *kay* | *mep* | *sap* |
| | | | *fip* | *kiv* | *mot* | *taz* |
| *pob* | - | *zad* | *gak* | *kug* | *naj* | *vup* |
| | | | *gol* | *lar* | *raz* | *yun* |

(rather than bisyllabic 'X' as in the original Misyak et al. study [20]) to more closely mimic natural language, where nonadjacent dependencies may not be denoted by perceptual cues. All sequences took the form 'AXB', however for the ungrammatical testing sequences the 'A' and 'B' elements did not match (e.g., $A_1XB_2$, *'bek kiv **tef**'* instead of *'bek kiv **jat**'*).

On each trial, participants were presented with both auditory and visual nonsense word stimuli (see Procedure). The auditory stimuli were naturally spoken by a female native English speaker recorded with an Edirol R-09HR (Roland) sound recorder and their amplitude was root-mean-square (RMS) balanced. The stimuli were presented at 70dB. The duration of all nonsense word stimuli was 600ms. The visual stimuli were presented on a computer monitor in white on a black background in courier font at font size 48 within a 300 by 300-pixel square (Fig 1).

## Procedure

Participants completed the SRT-AGL task first, which consisted of several different blocks. First were 6 'Learning Blocks', in which participants were presented only with grammatical sequences. Every possible 'AXB' sequence was presented once per block for a total of 72 trials, and the order of the sequences was randomised. These blocks gave the participants the opportunity to implicitly learn the nonadjacent dependencies. Following the 6 Learning Blocks there was a 'Testing Block' of 24 trials, in which participants were presented with only ungrammatical sequences (e.g., $A_1XB_2$). In the Testing Block all ungrammatical pairs of 'A' and 'B' elements (i.e., $A_1\_B_2$, $A_1\_B_3$, $A_2\_B_1$, $A_2\_B_3$, $A_3\_B_1$, $A_3\_B_2$) were presented equally frequently and each 'X' element was presented once. Finally, there was a 'Recovery Block' of 72 trials consisting only of grammatical sequences. This block was identical to the first 6 Learning Blocks. There were no pauses or information screens separating the different blocks, so participants were unaware that they were transitioning to or from the Testing Block. There was one optional pause halfway through the experiment, between Learning Blocks 4 and 5, to give the participants an opportunity to take a short break, if they desired. There was no relationship between break duration and learning (see S1 Table in S1 File).

In all the blocks, after completing each trial participants were given a score from 0–100 based on their reaction times, to encourage them to respond as quickly as possible (1/sum of reaction times (in seconds) multiplied by 100). This score was based on their total reaction time across the whole sequence, not specifically to the final, predictable 'B' element. This was to avoid any possibility that participants might preferentially respond quickly only to certain elements in the sequence to receive a higher score.

Following the SRT-AGL task, participants took part in the Sequence Completion and Grammaticality Judgement tasks (see General Methods).

## Data analysis

In the SRT-AGL task, participants completed 6 Learning Blocks, containing grammatical sequences only, followed by an ungrammatical Testing Block, followed by a final grammatical Recovery Block. We conducted repeated measures ANOVAs to compare reaction times across blocks. As in previous studies using this methodology, if learning had occurred during the SRT-AGL task, then we would predict faster reaction times to the final, predictable 'B' element ($RT_{A-B}$) in Learning Block 6 compared to Learning Block 1, as well as faster reaction times in Learning Block 6 compared to the Testing Block and faster reaction times in the Recovery Block compared to the Testing Block. Therefore, we also conducted planned comparisons using paired sample *t*-tests to compare reaction time differences between these blocks of the SRT-AGL task. Performance in the Sequence Completion and Grammaticality Judgement task

was compared to chance (50%) using one sample $t$-tests. To correlate performance on the implicit SRT-AGL task with the more explicit tasks, Pearson's correlation coefficients were calculated.

## Results

In this experiment we used an SRT-AGL task to assess the implicit learning of nonadjacent dependencies between nonsense words, based on the paradigm developed by Misyak et al. [20]. Misyak and colleagues found that participants got faster at responding to predictable 'B' elements over the course of the experiment ($RT_{A-B}$ increased from Learning Block 1 to Learning Block 6), and that responses slowed when presented with ungrammatical sequences in the Testing Block (Block 7), before increasing again in the Recovery Block (Block 8). We predicted similar effects in this experiment; however, the results were less clear.

We conducted a repeated measures ANOVA (with Greenhouse-Geisser corrections) with Block (8 blocks) as a within-subjects factor. There was no significant main effect of Block ($F_{2.49,67.29}$ = 2.757, $p$ = .059, $\eta^2_p$ = 0.093, 90% CIs = [0, 0.189]). Planned comparisons indicate that there was no increase in the reaction time difference ($RT_{A-B}$) from the first Learning Block to the final Learning Block, (paired-sample $t$-tests; $t_{27}$ = 0.138, $p$ = 0.89, $d$ = 0.026, 95% CI = [-0.345, 0.396]; Fig 2A, see S2 Table in S1 File for descriptive statistics), showing that the participants did not get faster at responding to the predictable stimuli over the course of the experiment. After completing the 6 Learning Blocks during which participants were only presented with grammatical sequences containing the predictable 'A-B' transitions, participants completed the Testing Block, in which all the sequences ended with an unexpected 'B' element (e.g., $A_1XB_2$). We compared response times in the final Learning Block with this Testing Block and found a trend towards slower responses ($t_{27}$ = 2.00, $p$ = 0.056, $d$ = 0.378, 95% CI = [-0.009, 0.759]), and to the subsequent Recovery Block where we found no significant differences ($t_{27}$ = -1.48, $p$ = 0.15, $d$ = -0.279, 95% CI = [-0.655, 0.101]). While the pattern of results is visually somewhat similar to the findings of previous studies using a similar paradigm [20], our data provide little evidence of implicit learning of the nonadjacent dependencies. Note, one potential explanation for the lack of an effect using the $RT_{A-B}$ measure may be that the task was sufficiently simple that participants' reaction times showed a floor effect, whereby it was not possible to respond any faster. We therefore calculated average reaction times to the 'A', 'X' and 'B' elements for each block of the experiment (see S2 Fig in S1 File). Average reaction times exceeded 800ms in all cases, suggesting that the lack of any differences in reaction times to the 'A' and 'B' elements was not driven by a floor effect. Moreover, reaction times to each element type were highly consistent (Cronbach's alpha reliabilities > 0.9 in all cases).

Following the SRT-AGL task, participants took part in the Sequence Completion task and the Grammaticality Judgment task. Performance was at chance levels in both the Sequence Completion task ($t_{27}$ = -0.22, $p$ = 0.83, $d$ = -0.042, 95% CI = [-0.412, 0.329]; Fig 2B) and the Grammaticality Judgement task ($t_{27}$ = 0.97, $p$ = 0.34, $d$ = 0.182, 95% CI = [-0.193, 0.554]; Fig 2C), again failing to demonstrate learning of the nonadjacent dependencies.

While at a group level we see little evidence of implicit learning, both the sequence completion task and grammaticality judgement task reveal a bimodal distribution of responses, with some participants performing at very high levels (green circles in Fig 2B and 2C) while the majority do not differ from chance (red circles). We categorised learners as participants who performed significantly above chance using binomial tests based on performance on the explicit tasks. The same three participants performed above chance on both the Grammaticality Judgment task and the Sequence Completion task. Interestingly, although only three participants fell into this category (rendering statistical analyses impossible), they all showed the

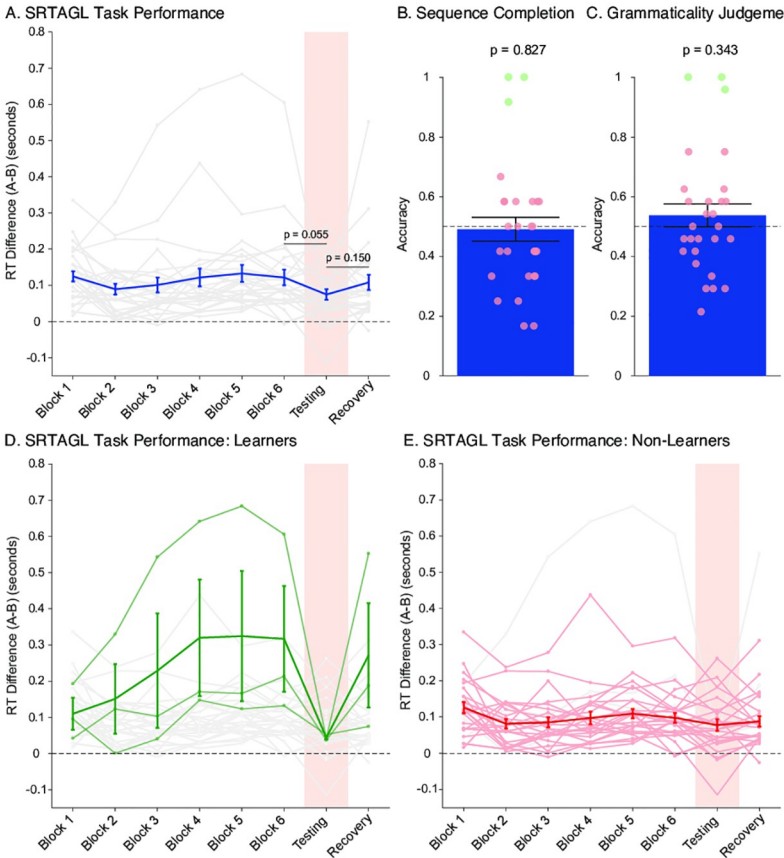

**Fig 2. Experiment 1 results.** A) Mean reaction time differences ($RT_{A-B} \pm$ SEM, thick blue line) for all participants for the 6 Learning Blocks, the Testing Block containing ungrammatical sequences (highlighted in pink), and the Recovery Block. Individual data is shown in grey. B) Mean ($\pm$ SEM) performance on the sequence completion task. Individual performance is shown in circles. Good learners are shown in green (based on individual binomial tests, $p < 0.05$), non-learners are shown in red ($p > 0.05$). C) Mean performance ($\pm$ SEM) on the grammaticality judgement task, including learners and non-learners, as in B. D) Mean reaction time differences ($RT_{A-B} \pm$ SEM) highlighting the good learners (green) based on the sequence completion and grammaticality judgement tasks, showing the predicted pattern of responses. E) Mean reaction time differences ($RT_{A-B} \pm$ SEM) for the non-learners (red) on the sequence completion and grammaticality judgement tasks, showing no learning on the SRT-AGL task.

patterns of performance on the prior SRT-AGL task that we predicted (Fig 2D). However, despite this, the majority of participants showed no evidence of learning in the SRT-AGL task (Fig 2E), or in the Sequence Completion or Grammaticality Judgement tasks.

Although based on good learning in only three participants, it is interesting that performance on the Sequence Completion task and Grammaticality Judgement task corresponds well with the implicit learning measured by the SRT-AGL task. This may suggest that these few participants had some conscious awareness of the regularities that they had learned during the SRT-AGL task. To directly assess the relationship between our implicit and explicit measures of learning, we calculated composite measures of SRT-AGL performance (based on the mean of the key measures of implicit learning: the mean of the difference between Learning Block 6 and Learning Block 1; Learning Block 6 vs Testing Block; Recovery Block vs Testing Block), and explicit task performance (based on the mean performance on the Sequence Completion task and Grammaticality Judgement Task). These two measures were highly correlated ($r = 0.71$, $p < 0.001$). However, a substantial amount of this effect was driven by the three good

learners and removing them from the analysis reduces the correlation substantially ($r = 0.47$, $p = 0.02$). However, given that no learning occurred in the SRT-AGL task in the majority of the participants, any correlations are unlikely to be informative, as they mostly reflect noise in the learning data.

At a group level, these results provide no clear evidence of nonadjacent dependency learning using either processing-based measures (SRT-AGL task) or reflection-based tasks (Sequence Completion and Grammaticality Judgement tasks). Evidence of learning of the nonadjacent dependencies was found in only a small minority of the participants, and while the correlation between implicit and explicit performance in these participants is intriguing, the majority of the sample showed no evidence of learning in either the SRT-AGL or subsequent Sequence Completion and Grammaticality Judgement tasks.

## Discussion

In Experiment 1, we found no evidence of learning across the SRT-AGL task at a group level. Participants' reaction times to the predictable 'B' elements did not decrease across grammatical blocks, and there was no difference in reaction times to these predictable 'B' elements between the grammatical and ungrammatical blocks. In the subsequent Sequence Completion and Grammaticality Judgement task, group performance was at chance levels. However, despite not finding evidence of learning at a group level, we did see very high levels of performance in the sequence completion and grammaticality judgement tasks in a small subset of participants (3 out of 28). These three participants also performed as predicted in the implicit SRT-AGL task. This bimodal distribution of performance is somewhat surprising, suggesting an almost binary distinction between those who learn the dependencies (most likely explicitly) and perform well on all the tasks, compared to those who show no measurable learning in either the implicit or more explicit tasks. Although the learners were defined based on their performance in the explicit tasks, these participants also showed learning in the prior SRT-AGL task. This suggests that these learners had acquired some knowledge that benefitted them in the SRT-AGL task, and which they could access more explicitly in the subsequent Sequence Completion and Grammaticality Judgement tasks. We do not highlight this pattern of results to suggest that this experiment successfully elicited learning. Rather, we aim to highlight the opposite point, that even if our results had been significant at the group level, had this bimodal pattern of results persisted we would be unwilling to claim that we found evidence for learning at the population level, when the majority of the population actually showed no such effect. When a group level effect is driven by a small number of (outlier) participants, we believe it is prudent to be sceptical of this task as an effective measure of learning.

At a group level we saw no evidence of learning in the Sequence Completion and Grammaticality Judgement tasks, even though such tasks have been used successfully in more traditional AGL paradigms. These AGL experiments typically consist of an exposure phase, where participants are exposed to grammatical sequences, followed by testing using a Grammaticality Judgement task. In the exposure phase, participants are instructed to attend to the stimuli with no distractions, to provide an opportunity for learning to take place. In the present experiment there was no passive exposure phase, instead participants were exposed to the dependencies during the SRT-AGL task's learning blocks. It is possible that the design of the task—in which participants attended to an auditory cue, identified its visual counterpart, then selected it, before moving on to the next element in the sequence–resulted in participants processing each element individually, rather than processing the sequence as a whole, as they might during a passive exposure phase. This may lead to the lack of learning we observed in this experiment (as we consider further in the General Discussion).

The findings from this experiment differed from those of Misyak et al [20], upon which this experiment was based. One other possible explanation for the conflicting results is the slight difference in the intervening 'X' elements used in this study: in Misyak et al.'s experiment, the 'X' elements were bisyllabic, meaning that they were perceptually distinct from the 'A' and 'B' elements, a difference which is thought to facilitate learning of the dependencies between the 'A' and 'B' elements [15]. In the current experiment, the 'A', 'X' and 'B' elements were all monosyllabic, meaning there were fewer perceptual cues that could be used to make the dependencies between the 'A' and 'B' elements more salient. While some previous research has suggested that nonadjacent dependencies are difficult for adults to learn without these additional cues (for a review, see [15]), many studies have shown successful learning of nonadjacent dependencies without using these bisyllabic stimuli (in adults: [47–49] and in infants: [50, 51]). Nevertheless, it is possible that the lack of any cues that differentiated the 'A', 'X', and 'B' elements could have made learning more difficult. Therefore, in Experiment 2, we used stimuli drawn from three different categories for the different element types.

In Experiment 1, we aimed to adapt Misyak et al. [20], to establish an effective processing-based measure of implicit statistical learning that could, in the future, be used to measure individual differences in learning, particularly in individuals with language difficulties, where nonsense words may not be appropriate. Given that this experiment failed to show implicit learning, we made several adaptations to the task, both to try and increase the chances that the task would elicit and measure implicit learning, and to make it more appropriate for individuals with language difficulties. Therefore, in Experiment 2 we aimed to improve the SRT-AGL task by using familiar objects as stimuli. We also introduced perceptual categories for the 'A', 'X' and 'B' elements with the goal of making the relationship between the 'A' and 'B' elements more salient. Furthermore, this modification removes the requirement for reading in the task, which makes it more appropriate when testing individuals with language difficulties.

## Experiment 2: Non-linguistic audio-visual SRT-AGL task

The key aim of these experiments was to develop a task, based on the original SRT-AGL paradigm [19, 20], that can be used to measure individual differences in implicit statistical learning, particularly in individuals with language difficulties. People with language difficulties like dyslexia often have problems reading, and particularly reading nonsense words that depend more heavily on phonological decoding [40]. Therefore, in this experiment we modified the task to remove the requirement to read the nonsense word stimuli, using familiar objects (i.e., animals, plants and food objects) instead of non-words. This allowed us to present auditory cues while the participant responded to visual stimuli, without requiring reading. Moreover, we used different categories of objects for the 'A', 'X' and 'B' stimuli, in an attempt to add perceptual cues that may facilitate learning (see General Discussion, and [15]).

The predictions for Experiment 2 remain the same as in Experiment 1: If the SRT-AGL task can effectively measure implicit statistical learning, then we would predict faster reaction times to the 'B' elements relative to the 'A' elements across Learning Blocks, followed by slower reaction times in the ungrammatical Testing Block, before restoration to faster reaction times in the final Recovery Block. Following the SRT-AGL task, participants completed the same Sequence Completion and Grammaticality Judgement tasks, using the new familiar object stimuli. As in Experiment 1, if learning had occurred and was consciously accessible, we would also predict that performance on the SRT-AGL tasks would correlate with performance during the subsequent, more explicit tasks. By contrast, if implicitly learned knowledge is inaccessible for conscious decision making, we would predict no correlation between the SRT-AGL task and the explicit tasks [43, 44].

## Methods

**Participants.** Thirty-eight adult participants (29 female, 7 male; mean age = 21.1) were recruited for Experiment 2. Sixteen participants were recruited using the Institute of Neuroscience participant pool at Newcastle University. An additional 22 participants were recruited through the Department of Psychology participant pool at Emory University. Testing methods and stimuli were identical regardless of testing location. All participants had normal or corrected to normal hearing and vision.

**Stimuli.** Experiment 2 used the same grammar as Experiment 1 (based on Gómez et al. [45]). To remove the requirement that participants must read the visual nonsense words, we replaced these stimuli with familiar objects drawn from the Bank of Standardized Stimuli Phase 2 [52] (Fig 3). As in Experiment 1, we used 3 'A', 3 'B' and 24 'X' elements, all represented by familiar objects. In this experiment, in an attempt to improve learning by providing additional cues to the categories of stimuli, all of the 'A' elements corresponded to plants, the 'B' elements corresponded to foods, and the 'X' elements corresponded to animals (see Fig 3). The auditory stimuli were the names of the images, naturally spoken by a female native English speaker recorded with an Edirol R-09HR (Roland) sound recorder and their amplitude was root-mean-square (RMS) balanced. The stimuli were presented at 70dB. The duration of all auditory stimuli was 600ms. The visual stimuli were presented at a size of 300 by 300-pixels, in colour on a white background, displayed against a black screen (Fig 3).

**Procedure.** Other than the new stimuli, the SRT-AGL task was identical to in Experiment 1, consisting of 6 Learning Blocks of 72 trials each, a Testing Block (24 trials) and finally a Recovery Block (72 trials). The sequences used in each block were designed and balanced identically to Experiment 1. The Sequence Completion Task and Grammaticality Judgement Tasks were also identical to that in Experiment 1, except using the new stimuli.

**Data analysis.** The data were analysed in the same way as in Experiment 1.

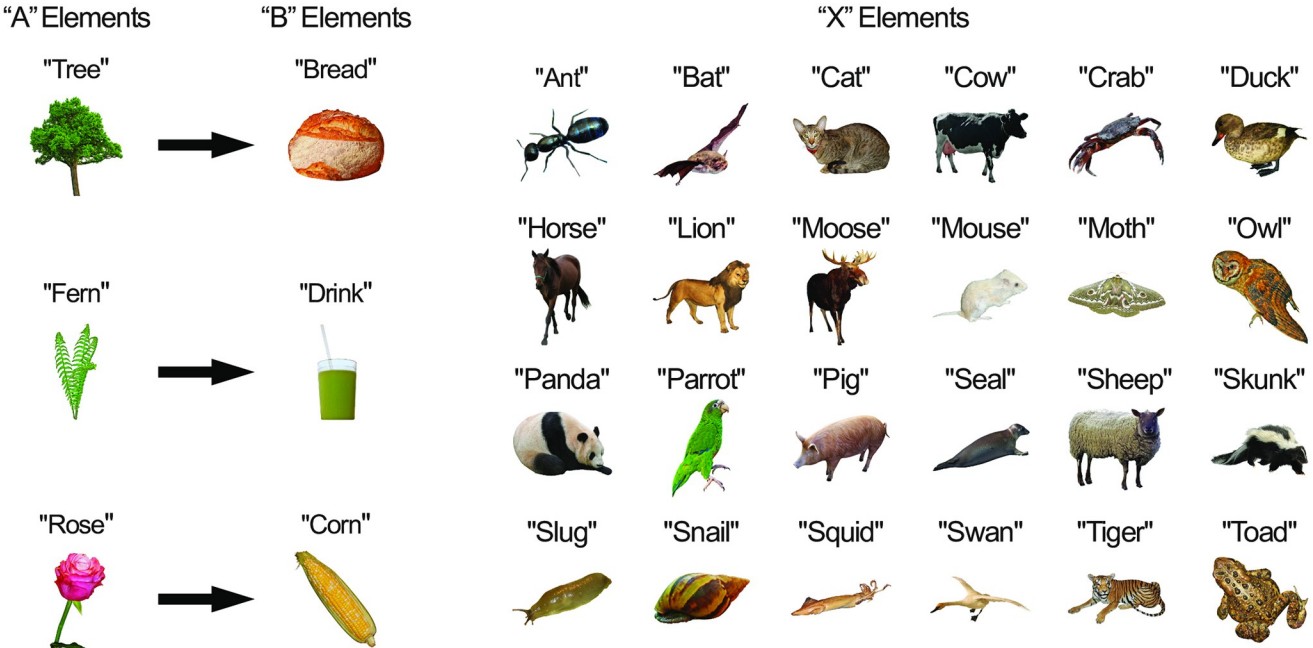

**Fig 3. Experiment 2 stimuli.** The 'A', 'B' and 'X' elements corresponded to plants, foods and animals respectively. The 'A' and 'B' elements always co-occurred, and sequences were presented in the form 'AXB' (e.g., 'tree cat bread'). There are 3 'A' and 'B' elements and 24 possible 'X' elements giving a total of 72 possible grammatical sequences. Each sequence was presented once in each Learning and Recovery Block (see Methods).

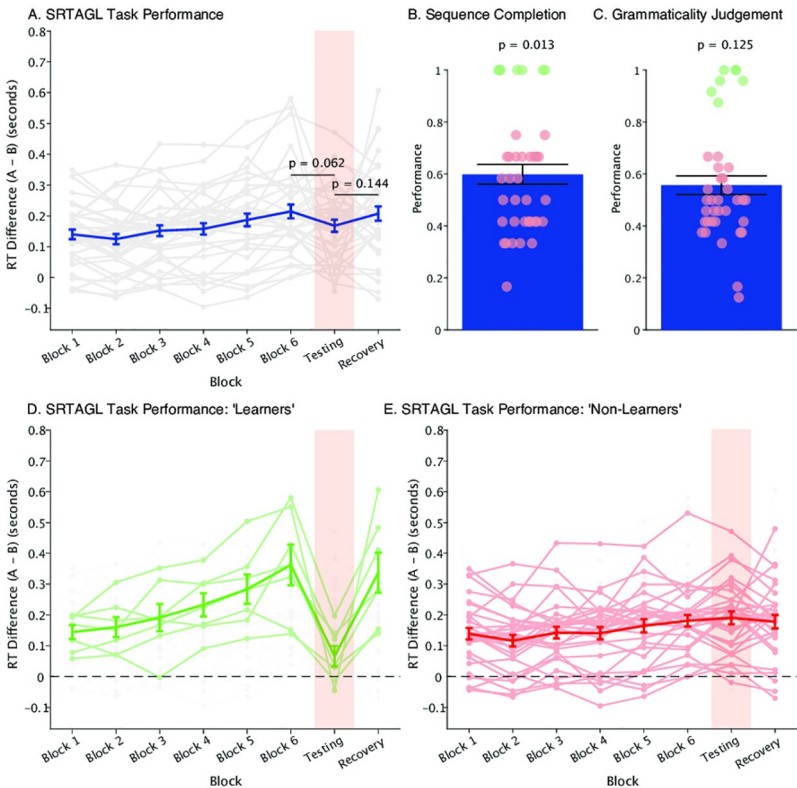

**Fig 4. Experiment 2 results.** A) Mean reaction time differences ($RT_{A-B}$ ± SEM, thick blue line) for the 6 Learning Blocks, the Testing Block containing ungrammatical sequences (highlighted in pink), and the Recovery Block. Individual data is shown in grey. B) Mean (± SEM) performance on the sequence completion task. Individual performance is shown in circles. Good learners are shown in green (based on individual binomial tests, $p < 0.05$), non-learners are shown in red ($p > 0.05$). C) Mean performance (± SEM) on the grammaticality judgement task, including learners and non-learners, as in B. D) Mean reaction time differences ($RT_{A-B}$ ± SEM) highlighting the good learners based on the sequence completion and grammaticality judgement tasks, showing the predicted pattern of responses. E) Mean reaction time differences ($RT_{A-B}$ ± SEM) for the non-learners on the sequence completion and grammaticality judgement tasks, showing no learning on the SRT-AGL task.

## Results

As in Experiment 1, we conducted a repeated measures ANOVA with Block (8 blocks) as a within-subjects factor to assess whether there were any changes in reaction time across blocks. We predicted that reaction times (measured as $RT_{A-B}$) would decrease from Learning Block 1 to Learning Block 6 as the participants implicitly learned the nonadjacent dependency between 'A' and 'B' elements, and that reaction times would increase in the Testing Block before decreasing again in the Recovery Block. There was a significant main effect of Block ($F_{3.52,130.07} = 7.212$, $p < .001$, $\eta^2_p = 0.163$, 90% CIs = [0.0616, 0.240]): planned comparisons showed participants' reaction times were faster in Learning Block 6 compared to Learning Block 1 ($t_{37} = -3.84$, $p < .001$, $d = -0.623$, 95% CIs = [-0.967, -0.271]; Fig 4A), showing that participants responded faster to the final 'B' elements at the end of the experiment. However, we saw no differences in reaction times between Learning Block 6 and the Testing Block ($t_{37} = 1.92$, $p = 0.062$, $d = -0.312$, 95% CIs = [-0.016, 0.636]) or the Testing Block and the Recovery Block ($t_{37} = -1.49$, $p = 0.145$, $d = -0.242$, 95% CIs = [-0.563, 0.083]). Therefore, again, these data do not provide convincing evidence that participants learned the dependencies in the SRT-AGL task.

In the Sequence Completion task, we observed performance somewhat above chance levels ($t_{37}$ = 2.582, $p$ = 0.014, $d$ = 0.419, 95% CIs = [0.084, 0.748]; Fig 4B), suggesting that some learning may have occurred. However, as in Experiment 1, there was a clear bimodal distribution of responses with some participants (5 out of 16) performing at ceiling levels, while the rest performed near chance. In the Grammaticality Judgement task group performance was no better than chance levels ($t_{37}$ = 1.57, $p$ = 0.125, $d$ = 0.255, 95% CIs = [-0.070, 0.576]; Fig 4C). However, the same five participants performed at very high levels (>90% correct). This pattern of results is very similar to those observed in Experiment 1, with the majority of participants showing no learning effects, while a subset of participants performed at very high levels. As in Experiment 1, we examined performance on the SRT-AGL task separately for these learners and non-learners based on their performance on the sequence completion and grammaticality judgement tasks. The 'learners' (7 out of 38 participants) showed the predicted pattern of learning in the SRT-AGL task (Fig 4D and 4E), mirroring the findings from Experiment 1. We again compared performance on composite measures of performance on the SRT-AGL task (the mean of the difference between Learning Block 6 and Learning Block 1; Learning Block 6 vs Testing Block; Recovery Block vs Testing Block) and the two more explicit tasks (mean of performance on the Sequence Completion and Grammaticality Judgement tasks). As in Experiment 1, these measures were highly correlated ($r$ = 0.69, $p$ = 0.003). However, this again driven by the seven participants who showed good learning, and when they were removed from the analysis the correlation disappeared ($r$ = -0.24, $p$ = 0.47). Once again, given that we found little evidence of learning, a lack of correlation between this performance and any other measure is unsurprising; if we were largely only measuring noise in the data, no correlations should be expected.

## Discussion

The results of Experiment 2 are very similar to those of Experiment 1, with no evidence of learning in the SRT-AGL task (no significant differences between Learning Block 6 and the Testing Block, or the Testing Block and the Recovery Block). Mixed evidence of learning was found for the explicit tasks, as group performance was above chance in the Sequence Completion task, but not above chance in the Grammaticality Judgement task. As in Experiment 1, any evidence of learning at a group level across the tasks was driven by a small subset of participants who performed very well across all the tasks.

In Experiment 2, we used new stimuli, familiar objects from three distinct categories, to provide additional cues to the 'A', 'X' and 'B' elements. However, despite these changes we found no convincing evidence of learning at the group level. This raises the question about whether the lack of effects in Experiments 1 and 2 are due to the stimuli we selected, or the design of the SRT-AGL task itself. Therefore, we designed a final experiment that contained several modifications, with the goal of trying to elicit implicit statistical learning.

In Experiment 3 we adapted the stimuli of the SRT-AGL task to avoid the requirement that participants read nonsense words (as in Experiment 1) or possess any prior linguistic knowledge (the names of objects in Experiment 2), by using abstract visual stimuli. As the abstract shapes do not have names, we used visual instead of auditory cues to indicate which stimuli the participants should choose. The participants were shown the target image in the centre of the screen, and the participants was asked to select the matching image. We also modified the design of the experiment so that instead of initially presenting several blocks of grammatical trials before a testing block, we now interspersed ungrammatical trials throughout each block, which may allow us to observe the trajectory of learning throughout the experiment. Finally, we also created two conditions: the first containing nonadjacent dependencies, as in

Experiments 1 and 2; and the second containing dependencies between adjacent stimuli. It is possible that the use of nonadjacent dependencies, which can be difficult to learn [15], may account for the poor performance across Experiments 1 and 2. Therefore, we included a condition with adjacent dependencies, which are typically easier to learn [6, 10, 12, 53], to ensure that the difficulty of the dependencies was not hindering performance in the SRT-AGL task.

## Experiment 3: Visual SRT-AGL task

Although beyond the scope of the current experiments, we hoped to design an SRT-AGL task that could in future be used to measure implicit statistical learning in individuals with language difficulties, such as dyslexia. In Experiment 3, in order to create a version of the SRT-AGL task that does not rely on processing auditory linguistic information, we created a completely visual version of the task, using abstract shapes as stimuli, and replacing auditory cues with visual cues. By using abstract shapes, the participants could not rely on knowing the names of objects, removing individual differences in familiarity with the object names.

We also revised the design of the SRT-AGL task: in the original [20] task, and in our Experiments 1 and 2, the SRT-AGL task comprised of 6 Learning Blocks, followed by one Testing Block, and one Recovery Block. Using this design, reaction times for grammatical and ungrammatical sequences cannot be compared until the testing block close to the end of the experiment. To assess the trajectory of learning across the SRT-AGL task, in Experiment 3 we implemented an oddball design, removing the Testing and Recovery Blocks and interspersing a small number of ungrammatical sequences (hereafter referred to as '*low frequency*' sequences due to the removal of discrete learning and testing phases), into each of the Learning Blocks containing primarily "grammatical" (*high frequency)* sequences.

We also introduced an additional reflection-based measure of learning, the 'Sequence Generation' task, which was completed after the Sequence Completion and Grammaticality Judgement tasks. In this task, participants were asked to create their own 3-element-long sequences that fit the same pattern they had seen previously. In this task, participants were not provided with any cues as to which stimuli to select. The Sequence Generation task was included to assess the extent to which any sequence knowledge that was obtained was available to consciousness, as the ability to generate and complete sequences would suggest more explicit knowledge of the structure [41, 42]. If the participants had conscious awareness of the dependencies, then performance on the Sequence Generation task should correlate with performance on the other tasks.

To ensure that poor performance in Experiments 1 and 2 could not be attributed to difficulties in learning nonadjacent dependencies, we also tested learning of adjacent dependencies. We also reduced the number of 'A' and 'B' elements in this task from three to two. This meant that both pairs of stimuli ($A_1$ and $B_1$, and $A_2$ and $B_2$) were always shown on the screen in each trial (one as the target and one as the foil). This change was made so that both dependencies could be displayed on the screen throughout the experiment, with the goal of increasing their salience and facilitating learning by helping participants track the relationships between these consistently presented elements.

We predicted faster reaction times to the 'B' elements relative to the 'A' elements across Learning Blocks for high frequency sequences, but not low frequency sequences. More specifically, we predicted that there would be an interaction between Learning Block and the grammaticality of the sequence: in the initial Learning Blocks, there would be no difference between reaction times for high and low frequency sequences, however as learning occurs, we predicted that reaction times for the predictable 'B' element relative to the unpredictable 'A' element would decrease for high frequency sequences, but not low frequency sequences. As in the

previous two experiments, we predicted above chance performance in the sequence completion and grammaticality judgement tasks. If learning is consciously accessible, we would expect participants who showed evidence of learning in the SRT-AGL task to also show above chance performance in the subsequent, more explicit Sequence Completion, Grammaticality Judgement and Sequence Generation tasks. If implicitly learned knowledge is not accessible for explicit processing, we would predict no correlation between the SRT-AGL task and the explicit tasks.

## Methods

**Participants.** 32 participants (23 female, 9 male; mean age: 21.86) were recruited using both the School of Psychology and Institute of Neuroscience participant pools at Newcastle University. 17 participants completed the adjacent condition, and 15 participants completed the nonadjacent condition.

**Stimuli.** We used 28 abstract shapes (2 'A', 2 'B' and 24 'X' stimuli) based on shapes from [5], shown in Fig 5. Unlike the Experiments 1 and 2, there were no auditory stimuli in this experiment, instead the abstract shapes were used as cues, with the participants being instructed to select the matching shape to the cue. The use of these abstract shapes meant that it was difficult to categorise the 'A', 'B' and 'X' stimuli in the same way as in Experiment 2, meaning there were no additional category-based cues in Experiment 3 that could facilitate learning.

**Procedure.** First, participants completed the visual SRT-AGL task, followed by the Sequence Completion, Grammaticality Judgement and Generation tasks.

**SRT-AGL task.** In the visual SRT-AGL task, participants completed 8 blocks of 24 trials, consisting of 20 high frequency and 4 low frequency sequences. As in Experiments 1 and 2, in

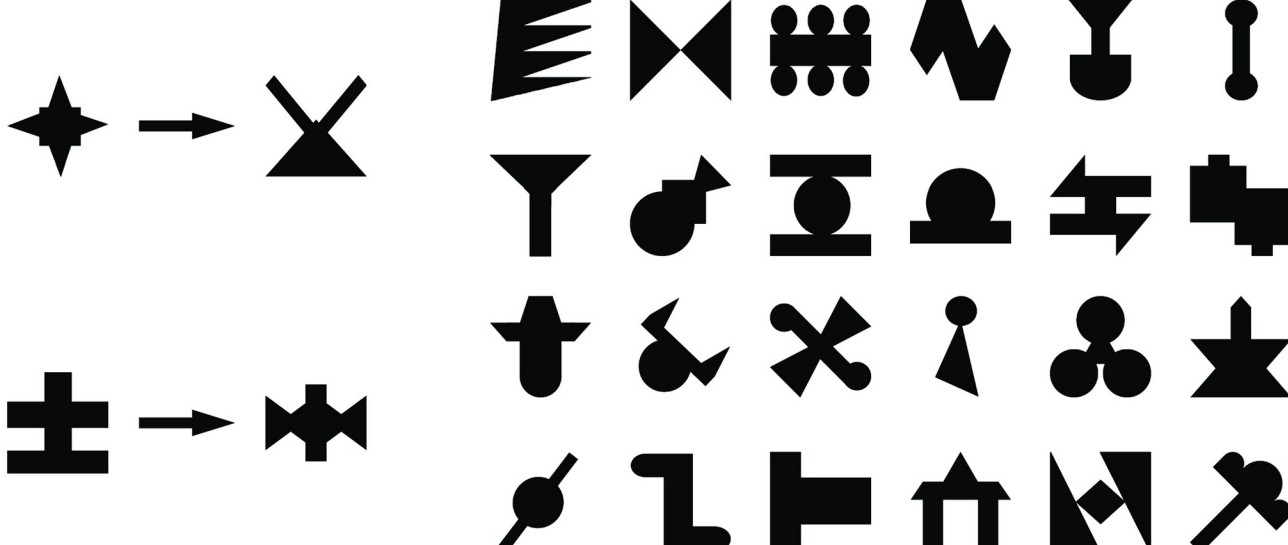

"A" Elements "B" Elements "X" Elements

**Fig 5. Experiment 3 stimuli.** The 'A' and 'B' elements always co-occurred, and sequences are presented in the form 'AXB' in the nonadjacent version of the task, and 'XAB' in the adjacent version. There are 2 'A' and 'B' elements, and 24 'X' elements. 24 sequences were presented per block, with each 'X' element presented once per Learning Block. 20 of the sequences in each block were high frequency (e.g., $A_1XB_1$), and for the 4 low frequency sequences, the dependencies between the 'A' and 'B' elements were switched around (e.g. $A_1XB_2$; $A_2XB_1$).

each trial, 6 abstract shapes (Fig 5) were presented on the screen (Fig 6). For each of the three horizontal positions in the sequence in turn, a cue stimulus matching one of the target shapes appeared between the targets after a 500ms delay (Fig 6). Participants were told to click on the matching shape as quickly as possible. The cue remained on the screen until the participant had selected a shape. As in Experiments 1 and 2, feedback on the speed of participant's responses was given after every trial, however in this experiment, happy or unhappy 'smiley faces' were presented on screen instead of scores. This was to remove the need to read any text during the task, which may not be suitable for any potential future testing with individuals with language difficulties. Participants were provided with the opportunity to take a break half-way through the experiment between blocks 5 and 6.

**Sequence Completion task.** The SRT-AGL task was immediately followed by the Sequence Completion task. As in Experiments 1 and 2, an array of 6 target elements was presented on the screen, and the first two cue elements were presented and responded to as normal. Instead of the final cue stimulus, the participants were presented with a question mark and asked to guess which target stimulus correctly completed the sequence (see Fig 6 and S3 Fig in S1 File). Participants completed 24 trials: each of the 'X' elements was presented once, and half of the sequences contained $A_1$ (with the correct response being $B_1$) and the other half of the sequences containing $A_2$ (with the correct response being $B_2$).

**Grammaticality Judgement task.** The Grammaticality Judgement task consisted of 24 trials in total, half of which were high and half low frequency sequences. In each trial, the entire visual sequence was presented simultaneously on screen. The method of response was identical to the previous two experiments, and the sequence remained on screen until the participant's response had been made.

**Sequence Generation task.** In this task participants were instructed to create 3 element long sequences, following the same pattern they had seen previously. In each of 24 trials, the participants were presented with 8 elements arranged in a circle on the screen: the 2 'A' and 'B' elements were always shown, as well as 4 randomly selected, non-repeating 'X' elements. Participants created their sequences by clicking on stimuli in order. Each trial was separated by a 500ms inter-trial interval. No feedback was given.

**Data analysis.** In the visual SRT-AGL task, for both the non-adjacent and adjacent conditions, we calculated reaction time differences: $RT_{Difference}$ = low frequency $RT_{A-B}$−high frequency $RT_{A-B}$. We used a repeated measures ANOVA to compare reaction time differences to high and low frequency sequences across blocks. Performance in the Sequence Completion and Grammaticality Judgement task was compared to chance (50%) using one sample *t*-tests. To correlate performance on the implicit SRT-AGL task with the more explicit tasks, Pearson's correlation coefficients were calculated.

## Results

In this experiment, there was no evidence of learning of the adjacent or nonadjacent dependencies across any of the tasks. We hypothesised that implicit learning would result in quicker responses to the predictable 'B' element than to the unpredictable 'A' element on high frequency trials relative to low frequency trials. The inclusion of high and low frequency sequences within every block of the SRT-AGL task differs from the SRT-AGL tasks in Experiments 1 and 2, which consisted of either grammatical or ungrammatical blocks. Therefore, within each block of the SRT-AGL task, for both the non-adjacent and adjacent conditions, we calculated reaction time differences: $RT_{Difference}$ = low frequency $RT_{A-B}$−high frequency $RT_{A-B}$. If learning had occurred, then this difference would increase across blocks. We conducted a 2x8 ANOVA, with Blocks (1 to 8) as within-subjects factors, and the Task

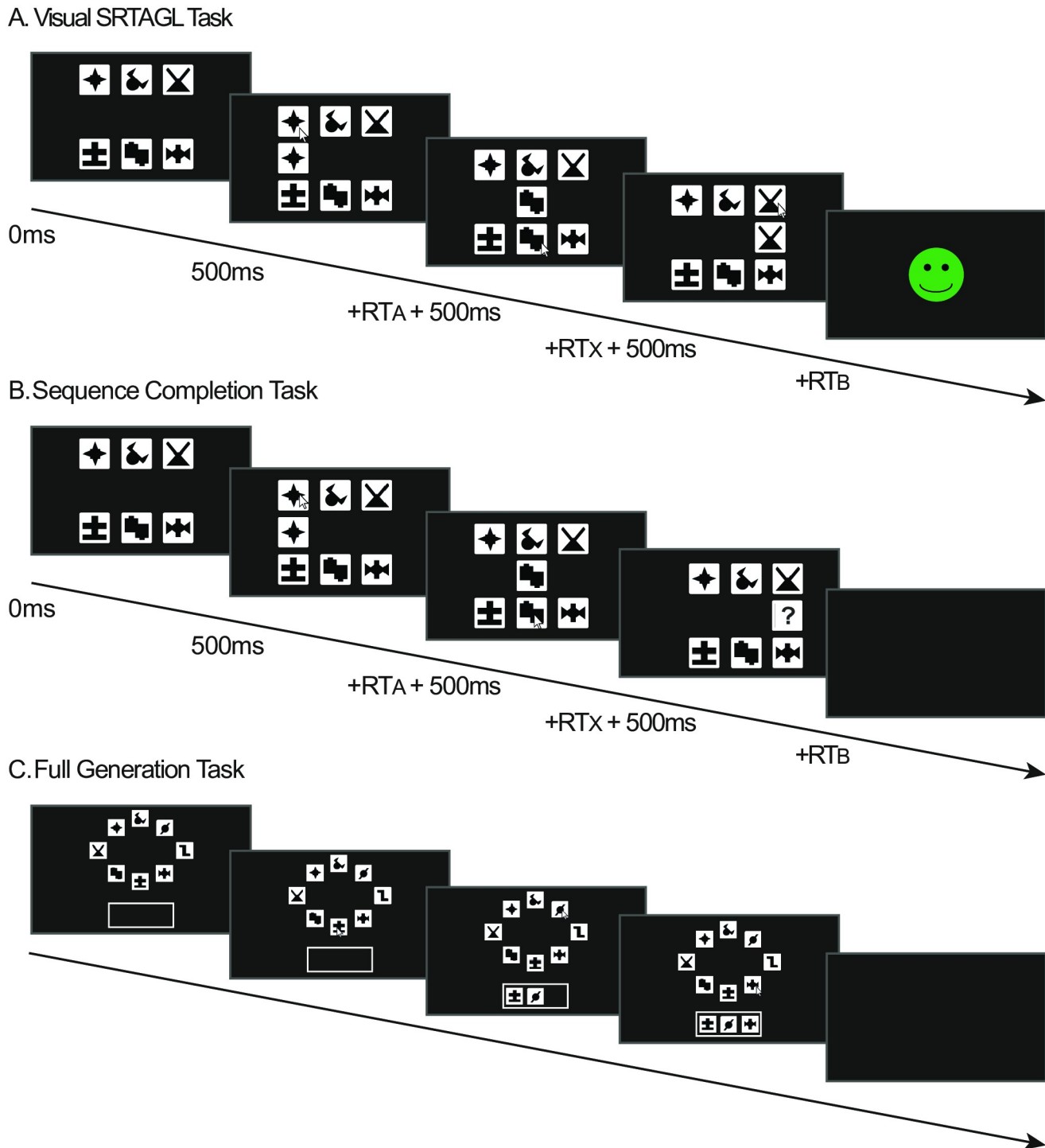

**Fig 6. Trial design for Experiment 3.** A) Visual SRT-AGL task showing the nonadjacent condition (for an example of the adjacent condition, see S3 Fig in S1 File). On each trial participants were presented with an array of 6 visual abstract shapes. In the nonadjacent condition, every trial contained 2 'A' elements in the left column, 2 randomly selected 'X' elements in the middle column, and 2 'B' items in the right column. In the adjacent condition, we presented 2 randomly 'X' elements in the left column, 2 'A' elements in the middle column, and 2 'B' elements in the right column. Participants were sequentially presented with a visual cue corresponding to one of the two visual stimuli in each column, and were asked to click on the matching visual stimulus as quickly as possible (see Methods). B) Sequence Completion task. As in the SRT-AGL task, participants were presented with a visual array of stimuli and responded to the first two visual cue stimuli as before. However, the final visual cue stimulus was replaced with a question mark, and the participants were asked to guess the shape that they felt completed the sequence. C) Generation task. In each of the 24 trials, the participants were presented with 8 elements arranged in a circle on the screen: the 2 'A' and 'B' elements were always shown, as well as 4 randomly selected, non-repeating 'X' elements. Participants created their sequences by clicking on the desired elements in order.

(nonadjacent or adjacent) as between subject factors. We found no significant effect of Block ($F_{7,210} = .498$, $p = .758$, $\eta^2_p = 0.016$, 90% CIs = [0, 0.016]) or Task ($F_{1,30} = .431$, $p = .516$, $\eta^2_p = 0.014$, 90% CIs = [0, 0.014]), or a Block*Task interaction ($F_{4,135} = 1.133$, $p = .345$, $\eta^2_p = 0.036$, 90% CIs = [0, 0.053]),)), which suggests that there were no differences in reaction times between the high and low frequency sequences in either the adjacent or nonadjacent conditions (see Fig 7). To determine whether reaction times were faster for high frequency over low frequency sequences across the experiment, we added an additional within-subjects factor of Grammaticality to the mixed ANOVA. There was no significant main effects (Grammaticality: $F_{1,30} = 0.024$, $p = .878$, $\eta^2_p = 0.00079$, 90% CIs = [0, 0.053]; Block: $F_{4.7,141.3} = .301$, $p = .953$, $\eta^2_p = 0.009$, 90% CIs = [0, 0.011]; Task: $F_{1,30} = 2.27$, $p = .143$, $\eta^2_p = 0.070$, 90% CIs = [0, 0.238]) or interactions between conditions (Grammaticality*Block: $F_{4.5,134.9} = .498$, $p = .758$, $\eta^2_p = 0.016$, 90% CIs = [0, 0.0311]; Grammaticality*Task: $F_{1,30} = .431$, $p = .516$, $\eta^2_p = 0.014$, 90% CIs = [0, 0.140]; Block*Task: $F_{4.7,141.3} = 1.18$, $p = .322$, $\eta^2_p = 0.038$, 90% CIs = [0, 0.0708]; Grammaticality*Block*Task: $F_{4.5, 134.9} = 1.13$, $p = .345$, $\eta^2_p = 0.036$, 90% CIs = [0, 0.0703]).

Participants also did not perform above chance in any of the explicit tasks. In the adjacent condition, participants performed slightly below chance on the Sequence Completion task ($t_{16} = -2.27$, $p = .037$, $d = -0.551$, 95% CI = [-1.055,-0.032]). In the subsequent Grammaticality Judgement task, participants performed at chance levels ($t_{16} = -.965$, $p = .348$, $d = -0.234$, 95% CI = [-0.713, 0.252]). In the nonadjacent condition, participants did not perform significantly above chance in either the Sequence Completion task ($t_{14} = 1.438$, $p = 0.172$, $d = 0.371$, 95% CI = [0.159, 0.889]) or Grammaticality Judgement task ($t_{14} = 1.00$, $p = .334$, $d = 0.258$, 95% CI = [-0.261, 0.769]). Unlike Experiments 1 and 2, we did not see a bimodal distribution in which some participants showed learning whereas others did not. Although two participants performed well in the nonadjacent Grammaticality Judgement task, this was not reflected in their performance across the SRT-AGL and Sequence Completion tasks.

Participants completed the Sequence Generation task to assess the extent to which any sequence knowledge that was obtained was available to conscious evaluation. As mentioned previously, the ability to generate and complete sequences would suggest more explicit knowledge of the structure [41, 42]. The Sequence Generation task required participants to create their own 3-element-long sequences based on the pattern they had seen previously. Although participants' performance was generally poor in this task, unlike the previous reflection-based tasks, there was no clear chance level that we could use to compare their performance to. Therefore, we examined whether participants who showed good performance on the explicit tasks were also more likely to produce grammatical sequences in the Sequence Generation task. We found no correlations between performance in the Sequence Generation task and performance in either the Sequence Completion (adjacent: $r = .28$, $p = .269$; nonadjacent: $r = .24$, $p = .395$) or Grammaticality Judgement (adjacent: $r = .16$, $p = .529$; nonadjacent: $r = .479$, $p = .071$) tasks. This was unsurprising given that only one participant showed evidence of learning across the experiment.

We also calculated composite measures of SRT-AGL performance based on the $RT_{(A-B)}$ difference between high and low frequency trials between Learning Block 1 and Learning Block 8, and explicit task performance (based on the mean performance in the Sequence Completion, Grammaticality Judgement, and Sequence Generation tasks). These measures were not correlated in either the nonadjacent ($r = -.308$, $p = .264$) or adjacent conditions ($r = .122$, $p = .640$), although this is again not surprising given the lack of learning across all tasks.

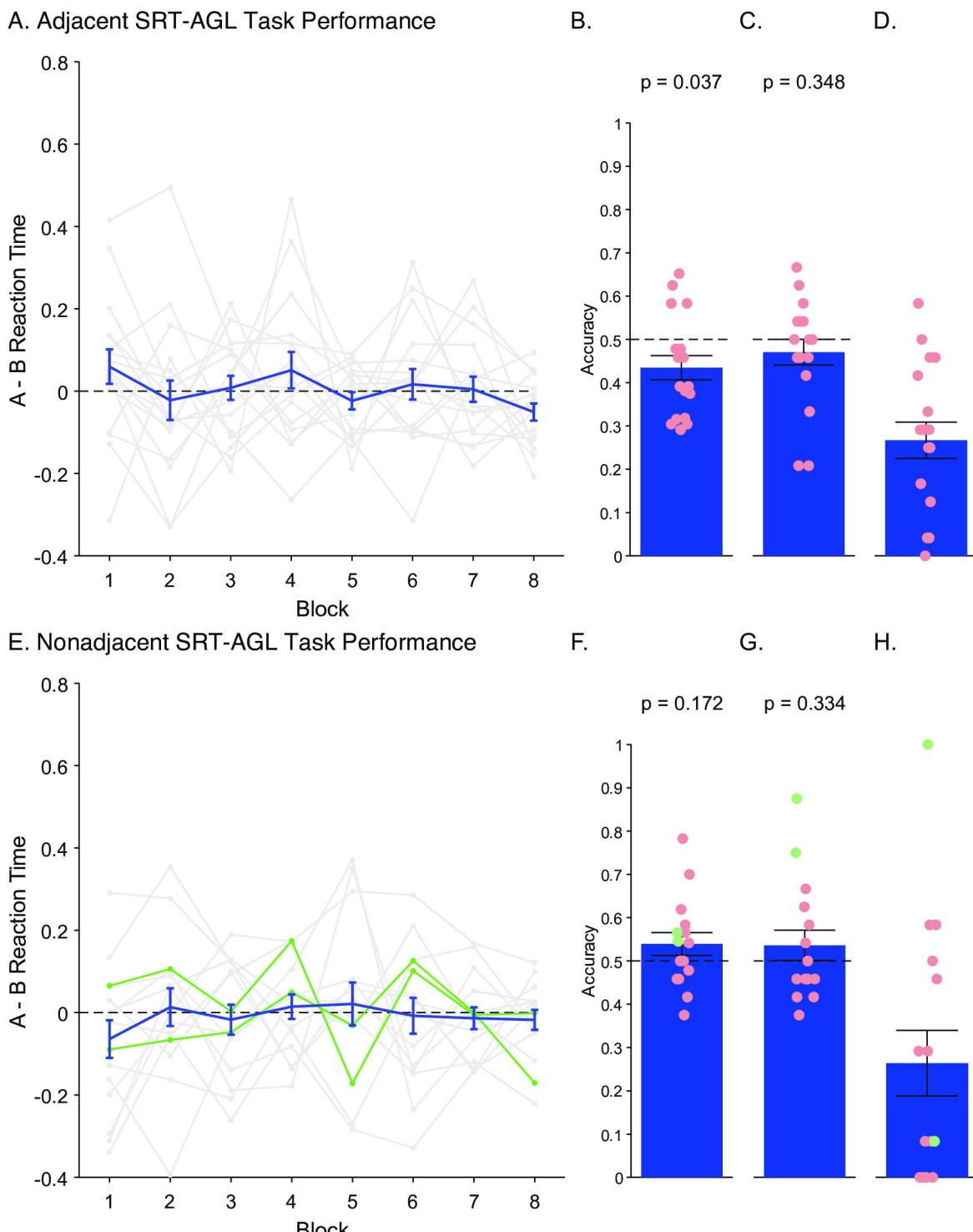

**Fig 7. Experiment 3 results.** Participants completed either the adjacent (A-D) or nonadjacent (E-H) task. Panels A and E show the difference in mean A—B reaction times ($RT_{difference} \pm SEM$) between high and low frequency sequences across blocks in the adjacent and nonadjacent tasks respectively. Individual data is shown in grey. Panels B and F show mean ($\pm$ SEM) performance in the Sequence Completion task in the adjacent and nonadjacent tasks respectively. Individual performance is shown in circles: 'good learners' are shown in green (based on individual binomial tests for the Grammaticality Judgement task, $p < 0.05$), non-learners are shown in red ($p > 0.05$). Panels C and G show mean performance ($\pm$ SEM) on the Grammaticality Judgement task in the adjacent and nonadjacent tasks respectively. Panels D and H show mean performance ($\pm$ SEM) in the Sequence Generation task in the adjacent and nonadjacent tasks respectively.

## Discussion

In this experiment we aimed to assess implicit statistical learning using a visual serial reaction time task that did not rely on any prior semantic or linguistic knowledge, with the hope of developing a task that could be used in other groups, including people with language difficulties. However, as in Experiments 1 and 2, we found no evidence of learning in the SRT-AGL task or the more explicit Sequence Completion, Grammaticality Judgement and Sequence Generation tasks, in either the nonadjacent or even very simple adjacent condition. Moreover, we only identified a single participant who showed good performance across the Sequence Completion, Grammaticality Judgement and Sequence Generation tasks, and this participant showed no evidence of learning in the visual SRT-AGL task. This may suggest that participating in the SRT-AGL task may inhibit learning that may have otherwise occurred during a standard AGL task.

We assessed learning using more explicit testing methods: the Sequence Completion, Grammaticality Judgement and Sequence Generation tasks. As in Experiments 1 and 2, there was no evidence of learning at the group level across these tasks in either the nonadjacent or adjacent condition. This could again be attributed to the lack of exposure phase that is typically found in artificial grammar learning paradigms. As in the audio-visual versions of the task (Experiments 1 and 2), completing the visual SRT-AGL task may act as a distraction, and draw attention away from the dependencies, resulting in a lack of learning and therefore poor performance across tasks. The lack of nonadjacent dependency learning in this task could be attributed to the more general inconsistency of learning of these dependencies reported in the literature (for a review, see [15]). However, lack of learning of simple adjacent relationships, particularly in the Grammaticality Judgement task, is unexpected, as previous research using these tasks has shown that adjacent relationships are typically learned readily without any additional cues [6, 10, 12, 53]. These findings strongly suggest that the complexity of nonadjacent relationships is not responsible for the lack of learning across these experiments. Instead, as outlined above, completing the SRT-AGL task itself may have inhibited learning of both types of dependency.

Experiment 3 showed no evidence of learning in a visual SRT-AGL task, or in subsequent explicit tasks, mirroring the findings of the previous two experiments. We made changes to the design of the SRT-AGL task to measure the trajectory of learning of the dependencies, which failed. The fact that participants did not learn the simple adjacent dependencies as well as the more complex nonadjacent dependencies suggests that the SRT-AGL paradigm failed to induce or measure learning of even simple relationships, as we discuss further in the General Discussion.

## General discussion

Implicit statistical learning has been studied using a wide range of paradigms over the course of the last 50 years [6, 12, 54]. However, many of these studies directly asked participants to categorise stimuli based on their grammaticality or 'well-formedness'. It is possible that these reflection-based measures might not capture some of the important aspects of implicit learning, which could be more accurately measured using more implicit, processing-based measures of learning. Therefore, in three experiments, we aimed to assess artificial grammar learning implicitly, using reaction time data, as well as more explicitly using both novel and more traditional assessments of implicit learning.

In all three experiments, we found no evidence of learning in either the SRT-AGL task or more explicit Grammaticality Judgement, Sequence Completion and Sequence Generation tasks. Additionally, as the majority of the participants in each experiment showed no evidence

of learning across any of the tasks, it is difficult to draw conclusions as to whether the tasks measure similar processes.

At a group level, the data from all three experiments support the null hypothesis that implicit statistical learning of the dependencies did not occur. However, at an individual level, there is evidence that a small number of participants did learn the grammar and performed very well across all tasks. Moreover, in Experiments 1 and 2, their performance on the processing-based SRT-AGL task and the subsequent reflection-based tasks were highly correlated. This suggests that in Experiments 1 and 2, we do not see evidence of learning in the processing-based tasks without also seeing evidence of learning in the reflection-based tasks, which may suggest that participants explicitly learned the dependencies during the SRT-AGL task, or that they did learn this information implicitly but were then subsequently able to explicitly access this during the explicit tasks. However, the results from these experiments are not able to separate these two possibilities. In either case, the vast majority of our participants failed to show any learning across all of the tasks. Therefore, we cannot conclude that this is an effective measure of implicit statistical learning, or that it is suitable for characterization of individual differences in implicit statistical learning in people with or without language difficulties.

SRT-AGL tasks aim to use reaction times to provide a processing-based measure of implicit statistical learning which is not confounded by spatial information. Previous research has shown that they can be effective measures of implicit statistical learning [19, 20, 46], and in our experiments we aimed to adapt the task to be more suitable for testing individuals with language difficulties in the future. We failed to show learning at a group level in all three experiments, despite a small number of participants showing predicted patterns of learning. The findings from these experiments raise the question of why these SRT-AGL tasks did not induce implicit statistical learning for most individuals. In traditional serial reaction time (SRT) tasks, participants are required to respond based on spatial positions [24], where one location predicts another, which relies on motor learning [27, 55, 56]. However, in SRT-AGL tasks, whilst the participants must still locate the elements on the screen, it is the stimuli that predict one another, not the location. This design was to ensure that participants learned the relationships between the stimuli, however it also meant that the participants only need to attend to the shapes enough to identify their key features, which could be less salient. It is possible that matching auditory or visual cues to their respective visual stimuli in the SRT-AGL task did not cause participants to take notice of the dependencies between stimuli, resulting in a lack of learning in the SRT-AGL tasks compared to more traditional SRT tasks. Specifically, the design of this task required that participants attend to an auditory or visual cue, then identify the appropriate target, then select it, before moving on to the next element in the sequence. It is possible that this design caused them to attend to each element of the sequence separately, rather than processing the whole sequence holistically, which might be more likely in a passive exposure condition. Therefore, attention to each individual element of the sequence may have hindered learning of the relationships *between* the elements, leading to the lack of learning of either adjacent or nonadjacent dependencies observed here. We note that SRT-AGL tasks have been used successfully in the past [19, 20], and we cannot fully account for why we found different results here. Rather, we note that despite three experiments and many modifications of the experimental design, we were unable to elicit learning via SRT-AGL tasks.

Although there was little evidence of learning across these three experiments, it is important to ensure that this data is available to contribute to the literature. Recent research has stressed the importance of publishing null results for the progression of science [57] and to avoid file drawer problems and publication bias [58], both of which have been reported within the implicit statistical learning literature [59–62]. Furthermore, in order to reveal more about the

key mechanisms underlying implicit statistical learning, it is important to understand which tasks do not induce or measure learning as well as those that do.

Identifying an implicit method of measuring implicit statistical learning abilities remains highly important, as many current measures of implicit statistical learning are primarily reflection-based, and therefore any attempt to measure the mechanisms underlying implicit statistical learning using such tasks may actually reflect explicit, conscious decision-making processes [18]. Although these SRT-AGL tasks find no evidence of learning, there is still a need for processing-based measures of implicit statistical learning that are not affected by conscious reflection. Recently, serial recall tasks have been used to measure implicit statistical learning more implicitly [21–23, 60]. As there was little evidence of implicit statistical learning across these SRT-AGL tasks, serial recall tasks may be a more useful processing-based measure that is able to provide a graded measure of performance over the course of the experiment, to better reflect the mechanisms underlying implicit statistical learning and reveal more about individual differences in these abilities.

## Supporting information

**S1 File.** This file contains all supporting information for this manuscript including: S1 Table: duration of breaks taken during experiment; S2 Table: descriptive statistics; S1 Fig: analyses of reaction times based on X element ($RT_{X-B}$); S2 Fig: reaction times for each element (A, X and B) for all experiments; S3 Fig: methods for Experiment 3, adjacent condition.
(DOCX)

## Acknowledgments

We thank Chris Petkov and Morten Christiansen for useful discussion.

## Author Contributions

**Conceptualization:** Holly E. Jenkins, Faye Smith, Nick Riches, Benjamin Wilson.

**Data curation:** Holly E. Jenkins, Phyllis Leung.

**Formal analysis:** Holly E. Jenkins, Benjamin Wilson.

**Methodology:** Holly E. Jenkins, Nick Riches.

**Writing – original draft:** Holly E. Jenkins, Benjamin Wilson.

**Writing – review & editing:** Holly E. Jenkins, Phyllis Leung, Faye Smith, Nick Riches, Benjamin Wilson.

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
