## [Decision Letter · Decision Letter 0]

16 Jan 2024

PONE-D-23-24683Assessing Processing-Based Measures of Implicit Statistical Learning: Three Serial Reaction Time Tasks Do Not Reveal Artificial Grammar LearningPLOS ONE

Dear Dr. Wilson,

Thank you for submitting your manuscript to PLOS ONE. After careful consideration, we feel that it has merit but does not fully meet PLOS ONE’s publication criteria as it currently stands. Therefore, we invite you to submit a revised version of the manuscript that addresses the points raised during the review process.

In light of the feedback from the reviews, you are recommended to follow the detailed suggestions provided by all the reviewers. In particular, all reviewers note the need to refine your study's methodology and clarify its theoretical underpinnings for a more impactful contribution. Focus on enhancing the robustness and explanation of your experimental design, ensuring it is systematically structured and aligned with established protocols. Additionally, a key area for improvement is in articulating a clear theoretical framework, particularly in distinguishing between implicit and explicit learning processes.

We look forward to receiving your revised manuscript.

Kind regards,

John Blake, PhD

Academic Editor

PLOS ONE

2.Note from Emily Chenette, Editor in Chief of PLOS ONE, and Iain Hrynaszkiewicz, Director of Open Research Solutions at PLOS: Did you know that depositing data in a repository is associated with up to a 25% citation advantage (https://doi.org/10.1371/journal.pone.0230416)? If you’ve not already done so, consider depositing your raw data in a repository to ensure your work is read, appreciated and cited by the largest possible audience. You’ll also earn an Accessible Data icon on your published paper if you deposit your data in any participating repository (https://plos.org/open-science/open-data/#accessible-data).

3. Thank you for stating the following in the Competing Interests/Financial Disclosure * (delete as necessary) section:

"The research was funded by a Sir Henry Wellcome Fellowship (BW: WT110198/Z/15/Z) and by ORIP/P51OD011132. (https://wellcome.org/). The funders had no role in study design, data collection and analysis, decision to publish, or preparation of the manuscript."

We note that you received funding from a commercial source: "Wellcome Trust"

4. Please note that your Data Availability Statement is currently missing  a direct link to access each database. If your manuscript is accepted for publication, you will be asked to provide these details on a very short timeline. We therefore suggest that you provide this information now, though we will not hold up the peer review process if you are unable.

 5. PLOS requires an ORCID iD for the corresponding author in Editorial Manager on papers submitted after December 6th, 2016. Please ensure that you have an ORCID iD and that it is validated in Editorial Manager. To do this, go to ‘Update my Information’ (in the upper left-hand corner of the main menu), and click on the Fetch/Validate link next to the ORCID field. This will take you to the ORCID site and allow you to create a new iD or authenticate a pre-existing iD in Editorial Manager. Please see the following video for instructions on linking an ORCID iD to your Editorial Manager account: https://www.youtube.com/watch?v=_xcclfuvtxQ.

Reviewers' comments:

Reviewer's Responses to Questions

**Comments to the Author**

1. Is the manuscript technically sound, and do the data support the conclusions?

Reviewer #1: Partly

Reviewer #2: Partly

Reviewer #3: Partly

2. Has the statistical analysis been performed appropriately and rigorously? 

Reviewer #1: N/A

Reviewer #2: Yes

Reviewer #3: Yes

3. Have the authors made all data underlying the findings in their manuscript fully available?

Reviewer #1: Yes

Reviewer #2: Yes

Reviewer #3: Yes

4. Is the manuscript presented in an intelligible fashion and written in standard English?

Reviewer #1: Yes

Reviewer #2: Yes

Reviewer #3: Yes

5. Review Comments to the Author

Reviewer #1: Review of “Assessing processing-based measures of implicit statistical learning: Three serial reaction time tasks do not reveal artificial grammar learning”

MS: PONE-D-23-24683

This paper examined an implicit learning of nonadjacent (in Experiments 1 and 2) and adjacent (between-subjects condition in Experiment 3) statistical regularities in three experiments using an AGL paradigm and a SRT task. In Experiment 1 (auditory stimuli), Experiment 2 (object stimuli with semantic information), and Experiment 3 (nonsense object stimuli), the authors did not observe the results suggesting implicit statistical learning, except for the sequence completion task in Experiment 2. The authors discussed the differences from the results of previous studies using the AGL paradigm in terms of the differences in the experimental paradigm (e.g., no familiarization/learning phase, number of syllables in X elements, response to spatial locations etc.).

The manuscript was interesting and the experimental results were clear. However, I have several concerns about the manuscript that should be addressed before the paper is published.

Major issue:

The authors pointed out the reproducibility of implicit statistical learning, however, did not describe in detail how this study was conducted. For example, I guess Experiment 1 used nonsense word and auditory stimuli, yet Figure 1 displayed nonsense object stimuli, such as those used in Experiment 3. I guess Figure 1 is a copy of Figure 6, and this is an error. Although Experiment 3 examined learning of both adjacent and nonadjacent regularities, Figure 6 shows only the examples of nonadjacent regularities; Figure 6 should also include examples of adjacent regularity condition.

In the SRT task, subjects were asked to click the same stimulus as the cue with the mouse, but it is unclear where the starting position of the mouse cursor was. Since the starting position of the mouse cursor is critical to RT to click the target, this point should be described in detail.

Moreover, an index of statistical learning in the SRT task is the difference of RT to B from that to A. This is also unclear. For example, because this detection task is so simple, the RT of most subjects could have shown a floor effect from Block 1. If this is true, then it is possible to point out that the SRT task used in this study simply failed to detect implicit statistical learning, and it is not true that implicit learning effect was not observed in the SRT task using the AGL paradigm. The authors should discuss this point as well (Not all readers will check the OSF data).

Furthermore, while this study discussed whether statistical learning is implicit or explicit, several previous studies on visual statistical learning have examined whether statistical learning is based on an implicit process (e.g., Bertels et al., 2012, JEPLMC; Cox et al., 2022, QJEP). They used subjective confidences (e.g., remember/know and post-decision wagering; see Persaud et al., 2007, Nat Neurosci) as well as a target detection task (Kim et al., 2009, Neurosci Lett; Rawal & Tseng, 2021, JEPHPP). The authors should discuss not only the relationship between these previous studies and this study, but also why the authors did not use measures such as subjective confidences.

The authors discussed for the lack of implicit/explicit statistical learning on the lack of exposure phase in the Discussion of Experiments 1 and 2, and on the lack of response to spatial locations in the General Discussion. Therefore, it is not clear for the readers why the current study could not replicate the results of previous studies. The authors should provide a consistent discussion as to why they could not replicate the results of previous studies. The authors should also discuss their results in relation to studies in which statistical learning was observed after categorical judgments in observing regularities (Turk-Browne et al., 2010, J Neurosci).

Minor points:

I think the authors should correct the title of this study. “Three serial reaction time tasks”: I would be mistaken by this title to think that this study used three types of serial reaction time tasks.

The sample size in this study differs across experiments. In particular, the sample size in Experiment 2 was smaller than in the other experiments, but learning effects were observed only in the sequence completion task in this experiment. The authors should describe how to decide the sample size in this study.

Effect sizes and confidence intervals should be described in the results. Because the conclusions of this study are based on null effects, the Bayes factor should also be described.

The authors discussed that implicit statistical learning was observed only in small sample size. However, the discussion based on the data of three subjects in Experiment 1, five in Experiment 2, and two in Experiment 3 is not necessary even for descriptive statistics because the sample sizes were very small.

The authors discussed the following between Experiment 1 and Experiment 2:

“However, it is possible that this could be attributed to differences in the ‘X’ stimuli between experiments. Misyak et al. used bisyllablic ‘X’ elements, which may have differentiated the ‘X’ elements from the ‘A’ and ‘B’ elements, and thus provided additional cues to the relationships between the monosyllabic ‘A’ and ‘B’ elements (Wilson et al., 2020). Therefore, to facilitate learning of the dependencies in Experiment 2, we provided additional cues to the categories of the stimuli…”

I think it is strange that the authors dealt with the number of syllables in X elements and categorical information of elements in the same way as cues for implicit statistical learning. They also discussed that categorical information in each element facilitates statistical learning. If this is true, they should cite some previous studies. Wilson et al. (2020) did not use visual objects with categorical information.

･ In these experiments… (Line 121): In this study?

･ we first aimed to replicate the original results (Misyak et al., 2009)… (Line 127): Why did the authors change the number of syllables in X elements?

･ In all three experiments…with language difficulties (Lines 162-184): I think these should be described in the General Discussion.

･ We used a repeated measures ANOVA was used to compare reaction time differences to grammatical and ungrammatical sequences across blocks? (Lines 753-755)

Reviewer #2: General comments:

As being familiar to the challenges of SL research, and especially measuring individual differences in SL, I was very happy to read a manuscript with null results, and I think the authors here set a good example by aiming to publish them. However, there are two aspects that make me concerned about the manuscript. First, the three studies are not systematically designed (although I could see why and how they follow from each other). For instance, I think that matching auditory and written form of nonwords, images and auditory labels of known objects and identical images involve different cognitive processes. Also, in Experiment 3, the measurement method also significantly changed. These do not necessarily pose a problem. But, and here comes my second concern, I think that the results are underinterpreted. The results from the three experiments point to the explanation that SRT-AGL tasks are not appropriate measurement methods for SL, and the authors also give some explanation related to the role of the perceptual discriminability of "X" elements and the role of attention. However, I missed a comprehensive discussion about why Misyak et al. (2009; 2010) could get significant learning compared to the present three experiments. In Experiment 1, the authors explain the null results of NAD learning with the lack of perceptual distinctness. However, in Experiment 2 and Experiment 3, this issue is eliminated (either by making "X" elements discriminable in Exp. 2, or by introducing adjacent dependencies in Exp. 3). Here, the authors shift to the explanation about the effect of attention in the task. To summarize, prior to publishing, I think that the manuscript needs a thorough, comprehensive (and maybe more unitary?) discussion about what could cause the null results compared to the original Misyak et al. results.

As a second general comment that might be helpful for the authors is related to their theoretical framing of SL. They separate implicit and explicit processes to discriminate more "pure" forms of SL and other processes. However, in my view, if we look at phenomenon-based definitions of SL (given that we lack an exact mechanism-level account), then the basic property of SL is that it is an unsupervised, bottom-up learning process (e.g., Conway, 2020), and implicitness is only a by-product, not a necessary property. Therefore, I'm not convinced either that the key to the reliable measurement of statistical learning is that we measure implicit representations, but rather that we exclude other cognitive processes theoretically independent from SL (e.g., executive functions, reasoning, etc.). Measuring explicit learning with "reflection-based" processes might also not be a perfectly proper approach, as implicitness does not necessarily exclude learning effects in these measures (moreover, it seems that scores on "processing-based" and "reflection-based" measures correlate more when learning is implicit, see Lukics & Lukács, 2021). Although, if the real aim of the manuscript is separating explicit vs. implicit processes, than the author's approach is proper, but I felt confused about the real aim of the studies (discriminating implicit vs. explicit processes, or discriminating "real" SL and other cognitive processes, and thus targeting SL abilities). This is not a fundamental issue with the manuscript, and I know that the levels of explanation are mixed in the literature, as well, (implicitness vs. explicitness and "pure" SL vs. the effect of other cognitive functions), however, this might be an interesting aspect for the authors to think about, and they should state the aim of the studies more explicitly.

Specific comments:

page 11, row 103: See also Lammertink et al. (2019; 2020), and Lukics & Lukács (2021) for other instances of SRT-like measures for SL, and Hunt & Aslin (2001) for an early adaptation of the SRT paradigm.

page 14, row 162: This paragraph would fit better in the General Discussion.

page 14, row 173: "Furthermore, file drawer problems...": For me, this seems to be again a matter for the discussion. Or did the authors hypothesize the null results originally?

page 15, row 186: I would not necessarily divide the General Methods section into subsections: it might be a bit misleading, and the reader would expect e.g. the exact stimuli in the Stimuli subsection. But this is just a comment, I will leave it to the authors!

page 15, row 201: I would mention the details of the ethical approval here and only refer back to it in the Participant subsections in the given experiment.

page 16, row 225: "Therefore...": Did the authors think about including unpredictable element ("X" element) RTs as a baseline? Do the authors know that the RTs for the initial "A" elements are reliable? Baseline RTs in participants (processing speed) should have a high reliability (> 0.8), and it might be worth assessing the reliability of "A" element RTs. I'm not sure about it, but I suspect that RTs for the initial elements are less reliable than those of the following elements, and might be more exposed to fluctuations, thus including more noise.

page 17, row 237: I think that all post-hoc tasks consisted of 24 trials, but it should be stated everywhere.

page 17, row 255: "were presented with sequences of three stimuli": Were these with novel or with old "X" elements, that is, did this require generalization?

page 24, row 404: "However...": A binomial test is a perfect statistical solution for discriminating learners from non-learners, but if participants really performed on chance level, then their performance in only noise, and there is absolutely no sense of a correlation between noise and implicit learning measures after removing learners. Maybe some participants "unlearned" dependencies during the Testing block and systematically chose the incorrect "B" responses for some given trials? It might be interesting to look at whether chance level performances are truly random on the individual level. However, given the results of Experiment 2 and Experiment 3, it can indeed be the case that this correlation is the result of mere chance.

page 27, row 440: "Misyak and Christiansen (2012)...": I recommend the Lammertink et al. (2019; 2020) papers. Maybe the SRT-AGL task used by Misyak et al. recruits more reasoning / hypothesis testing processes than e.g., the method of Lammertink et al.?

page 27, row 446: Do the authors think that this perceptual distinction promotes more explicit reasoning / hypothesis generation? This might be supported by the more emphasized bimodal distribution of the original Misyak et al. study, as well. Or does perceptual discrimination affect bottom-up, unsupervised learning as well? How infant studies support one solution or another? (This leads to a more general question, maybe worth discussing in the General Discussion: if such a level of perceptual discrimination is needed for NAD learning to take place, is this really SL according to the definition of SL (see Conway, 2020), or more likely reflect, e.g., reasoning / hypothesis testing abilities? But this is only a suggestion that the authors, of course, don't have to accept.)

page 28, row 461: "It is possible...": Do the authors think that it drew attention away from the unsupervised form of SL? Or again, is it a matter of intentional learning in the case of NAD learning?

page 31, row 508: including 16 participants in Experiment 2 compared to the sample sizes of Experiment 1 and Experiment 3 seems to be unjustified for me. Did the authors conduct a power analysis prior to the data collection?

page 31, row 525: "The duration of all nonsense word stimuli was 600ms.": Were these then auditory labels of objects or nonsense words?

page 47, row 859: "The fact that participants...": What mechanisms could cause this in the task? I already mentioned this in the general comments, but this is an instance where the authors should be more explicit about it (or in the General Discussion).

Minor comments:

page 9, row 53: "similar processes are involved"  "similar processes are assumed to be involved"

page 14, row 165: "However...": it is not clear how the two parts of this sentence follow from each other.

page 30, row 484: "instead using familiar objects (i.e., animals, plants and food objects) instead of non-words": first instance of "instead is unnecessary".

page 43, row 770: "AVOVA"  "ANOVA"

In general, the authors might would like to follow APA 7 formatting.

The link pointing to the OSF repository doesn't work for me, although I could find the repository by the title of the manusript.

References:

Hunt, R. H., & Aslin, R. N. (2001). Statistical learning in a serial reaction time task: access to separable statistical cues by individual learners. Journal of Experimental Psychology: General, 130(4), 658.

Lammertink, I., Van Witteloostuijn, M., Boersma, P., Wijnen, F., & Rispens, J. (2019). Auditory statistical learning in children: Novel insights from an online measure. Applied Psycholinguistics, 40(2), 279-302.

Lammertink, I., Boersma, P., Wijnen, F., & Rispens, J. (2020). Children with developmental language disorder have an auditory verbal statistical learning deficit: Evidence from an online measure. Language Learning, 70(1), 137-178.

Lukics, K. S., & Lukács, Á. (2021). Tracking statistical learning online: Word segmentation in a target detection task. Acta Psychologica, 215, 103271.

Reviewer #3: This paper reports weak to null findings for learning artificial grammars from SRT-based tasks, which were adapted from some prior studies (e.g., Misyak et al., 2009). The experiments were motivated by the need to identify learning measures that would better gauge implicit expressions of statistical learning across individuals, and it was hoped that RT processing-based measures from the adapted tasks would serve this role. In light of weak to no evidence for learning across their three experiments, the authors pin this failure on the SRT-AGL task methodology itself, concluding that the training tasks may have inhibited learning.

The methods and analyses are clearly described, and the experiments are well motivated within the literature. I have four primary comments:

(1)

A main limitation of the paper is that it doesn’t demonstrate that the grammars and stimuli are robustly learnable under other comparable conditions, leaving the specific effect of the task method (i.e., the “SRT-AGL”) unclear.

E1, in particular, adapts the nonadjacent grammar from Gómez (2002) that has been used across several studies in the statistical learning (SL) literature, but changes the syllable construction of the middle elements of the strings. The authors acknowledge this may have critically affected the results, and the referenced explanation from Wilson et al. 2020 seems plausible here. E2 and E3 introduce further changes in more visually-based versions, with E3 converting the grammar to one with adjacent dependencies for half the participants. (It is presumed, as a generalization, that adjacent dependencies are easier to learn, although this is not always the case and will depend on learning conditions.)

There is no mention of participants being able to otherwise learn (at the group-level) — either by the authors or elsewhere — from these specific altered grammars and training materials using other training tasks. One possibility is that the altered grammar presentations are relatively difficult to learn on their own merits, irrespective of the choice of task method. Another possibility is that the SRT-AGL implementation is harder to learn from (compared to a more traditional AGL task), which can result in null or weak findings for more difficult grammar iterations (as may be the case here) but not for more robust ones. Either possibility could account for the differences between the authors’ findings and those in the prior literature.

It would be informative to the authors’ claims if participants’ levels of learning from these specific grammar versions were ascertained using other AGL or SL tasks (ideally with the same stimuli using comparable amounts of training exposure, stimuli duration, etc.). For the grammars in E3, this would actually include exposure to ungrammatical strings during the training phase, as consistent with the approximately 17% of ungrammatical sequences presented in that training regime. I appreciate that this may entail using less ‘implicit’ assessment measures by the authors’ standards.

But in the absence of such evidence for learnability, the contribution (if any) of the SRT-AGL based task to produce this failure is unclear. The claims of the paper and the discussion should reflect this limited scope for conclusions.

(2)

Related to the above, E1 is not an exact replication of Misyak et al., 2009 — and seemingly not a close one given various differences, some of which may have been critical.

While the grammar may appear ostensibly the same in E1, the syllable structure of the middle elements was changed (as noted above); and different stimuli were necessarily used in implementing this change. As the authors concede that this change may have undermined learning, I won’t expand on this point here, other than to underscore that it is unclear whether, and to what extent, people readily learn the Gómez grammar with this modification.

Another potentially substantial difference relates to the inclusion of an optional mid-task break, which could be a source of variability across participants in reaction times for the later blocks, from which the test comparisons were derived. (The authors don’t report the proportion of participants who opted for the break, and whether the duration of the break was allowed to vary across participants, so more information in this regard may address the extent to which this concern is relevant.)

There are further differences in E1, though perhaps not vital ones, and of course, the task and measures become increasingly more altered across E2 and E3.

In light of such distinctions, would it be more appropriate to refer to the task more consistently as an adaptation, rather than replication? In any event, it may be helpful to readers unfamiliar with the original work to explicitly identify all the various differences in E1; this might also inform generalizability of the task to different parameters.

(3)

What is the scientific rationale for the reduced sample (16) in E2, relative to E1 (28)?

E2 does report evidence of learning on one of the three processing-based measures, and on one of the two post-processing measures. The remaining measures show trending or directionally consistent indications of learning.

However, the sample size seems small in this context, particularly for reaction time differences that may be expected to be small. Could this experiment have been underpowered to detect learning?

(E3 also has similarly small sample sizes.)

(4)

E3 alters the SRT-AGL task by removing the ‘testing’ block of ungrammatical sequences and replacing this with an oddball design where a small number of ungrammatical sequences are interspersed with the grammatical sequences throughout. RT metrics are then computed with respect to differences between responses for grammatical and ungrammatical sequences per block.

However, the proportion of ungrammatical sequences (1 out of 6) seems potentially high for artificial grammar training. At this rate, the ungrammatical sequences may be perceived as a lower-frequency regularity of the grammar (i.e., as consistent with the grammar) — with consequences, of course, for participants’ learning and their expression of that learning.

At the same time, the corresponding number of ungrammatical sequences (4) is low per block. Using such an extremely low number of items in the calculation of a difference score would be expected to detrimentally affect the reliability of that test measure.

Either or both of these aspects could be contributors to the null findings of E3. As in pt. (3) above, some attention to and discussion of these issues are warranted.

In sum, there are myriad issues that could have contributed to null findings across the authors’ experiments. As these are left unresolved, the authors should revise their discussion and claims accordingly to take these considerations into account.

As an aside:

Although I do believe these experiments have restricted scope and are inconclusive on the merits/deficits of the processing-based measures, per the points above, I am sympathetic to the concerns the authors have about these SRT-based and traditional AGL tasks in sensitively gauging implicit learning, especially at the level of individual differences for learning as it unfolds over time. There can be issues regarding the psychometric reliabilities of reaction times more generally, which perhaps these adapted tasks and online measures may have inherited. I don’t expect the authors to delve into these issues in their paper, but do appreciate the complexity of this matter and wish them success in future investigations.

Other points:

The wrong file appears to be uploaded for Fig.1. (Figures 1 and 6 are identical.)

Lines 111-112: “random sequences containing no predictable dependencies”  technically not random, as some grammatical structure was retained (the initial/middle/final positional status of nonsense words) and the predictive dependencies were systematically broken

Reference information for Conway 2005 and Misyak et al. 2009 are incomplete. Missing information for Pacton et al. 2015 citation.

Lines 265-266: for clarity, I assume this is the mean for each subject?

What are the RT means or mean differences for the reported results? While differences are presumably minimal to nil, these are hard to read from the manner in which the figures are scaled and the values are unreported in the text.

Lines 440-441: The bimodal distribution was reported for performance from a *standard AGL* task and grammaticality test measure. Those same authors’ report of performance on an analogous metric to the ‘Sequence Completion’ measure, elicited after using the AGL-SRT task, shows more of a continuum (Misyak et al., 2010, Topics in Cognitive Science).

Line 515: “Gomez et al. 2002”  Gómez 2002

For E1 and E2, how many items were in the Sequence Completion and Grammaticality Judgment tasks?

The Figure 6 caption should clarify that it illustrates the nonadjacent (and not adjacent) condition of the visual task.

Line 802: “adjacent Sequence Generation task”  Sequence Generation task

6. PLOS authors have the option to publish the peer review history of their article (what does this mean?). If published, this will include your full peer review and any attached files.

Reviewer #1: No

Reviewer #2: No

Reviewer #3: No

---

## [Decision Letter · Decision Letter 1]

29 Jul 2024

Assessing Processing-Based Measures of Implicit Statistical Learning: Three Serial Reaction Time Experiments Do Not Reveal Artificial Grammar Learning

PONE-D-23-24683R1

Dear Dr. Wilson,

We’re pleased to inform you that your manuscript has been judged scientifically suitable for publication and will be formally accepted for publication once it meets all outstanding technical requirements.

Kind regards,

John Blake, PhD

Academic Editor

PLOS ONE

Additional Editor Comments (optional):

Reviewers' comments:

Reviewer's Responses to Questions

**Comments to the Author**

1. If the authors have adequately addressed your comments raised in a previous round of review and you feel that this manuscript is now acceptable for publication, you may indicate that here to bypass the “Comments to the Author” section, enter your conflict of interest statement in the “Confidential to Editor” section, and submit your "Accept" recommendation.

Reviewer #1: All comments have been addressed

2. Is the manuscript technically sound, and do the data support the conclusions?

Reviewer #1: Yes

3. Has the statistical analysis been performed appropriately and rigorously? 

Reviewer #1: Yes

4. Have the authors made all data underlying the findings in their manuscript fully available?

Reviewer #1: Yes

5. Is the manuscript presented in an intelligible fashion and written in standard English?

Reviewer #1: Yes

6. Review Comments to the Author

Reviewer #1: The authors have addressed all of my concerns. The manuscript makes an interesting contribution to the literature.

7. PLOS authors have the option to publish the peer review history of their article (what does this mean?). If published, this will include your full peer review and any attached files.

Reviewer #1: No

---

## [Editor Report · Acceptance letter]

11 Sep 2024

PONE-D-23-24683R1 

PLOS ONE

Dear Dr. Wilson, 

I'm pleased to inform you that your manuscript has been deemed suitable for publication in PLOS ONE. Congratulations! Your manuscript is now being handed over to our production team.

Kind regards, 

on behalf of

Dr. John Blake 

Academic Editor

PLOS ONE